# Cytomegalovirus protein m154 perturbs the adaptor protein-1 compartment mediating broad-spectrum immune evasion

Ivana Strazic Geljic[1†], Paola Kucan Brlic[1†], Guillem Angulo[2], Ilija Brizic[1,3], Berislav Lisnic[1,3], Tina Jenus[1], Vanda Juranic Lisnic[1,3], Gian Pietro Pietri[1], Pablo Engel[2,4], Noa Kaynan[5], Jelena Zeleznjak[1,3], Peter Schu[6], Ofer Mandelboim[5], Astrid Krmpotic[3], Ana Angulo[2,4], Stipan Jonjic[1,3]*, Tihana Lenac Rovis[1,3]*

[1]Center for Proteomics, Faculty of Medicine, University of Rijeka, Rijeka, Croatia; [2]Immunology Unit, Department of Biomedical Sciences, Faculty of Medicine and Health Sciences, University of Barcelona, Barcelona, Spain; [3]Department of Histology and Embryology, Faculty of Medicine, University of Rijeka, Rijeka, Croatia; [4]Institut d'Investigacions Biomèdiques August Pi i Sunyer, Barcelona, Spain; [5]The Lautenberg Center for General and Tumor Immunology, The BioMedical Research Institute, Hadassah Medical School, The Hebrew University, Jerusalem, Israel; [6]Zentrum für Biochemie und Molekulare Zellbiologie Institut für Zellbiochemie, Georg-August-Universität Göttingen, Goettingen, Germany

*For correspondence:
stipan.jonjic@medri.uniri.hr (SJ);
tihana.lenac@medri.uniri.hr (TLR)

†These authors contributed equally to this work

**Abstract** Cytomegaloviruses (CMVs) are ubiquitous pathogens known to employ numerous immunoevasive strategies that significantly impair the ability of the immune system to eliminate the infected cells. Here, we report that the single mouse CMV (MCMV) protein, m154, downregulates multiple surface molecules involved in the activation and costimulation of the immune cells. We demonstrate that m154 uses its cytoplasmic tail motif, DD, to interfere with the adaptor protein-1 (AP-1) complex, implicated in intracellular protein sorting and packaging. As a consequence of the perturbed AP-1 sorting, m154 promotes lysosomal degradation of several proteins involved in T cell costimulation, thus impairing virus-specific CD8[+] T cell response and virus control in vivo. Additionally, we show that HCMV infection similarly interferes with the AP-1 complex. Altogether, we identify the robust mechanism employed by single viral immunomodulatory protein targeting a broad spectrum of cell surface molecules involved in the antiviral immune response.

## Introduction

Infection with human cytomegalovirus (HCMV; Human herpesvirus 5), highly prevalent in the world population, is usually asymptomatic in immunocompetent individuals and results in lifelong latency in the host organism. However, it is a significant cause of morbidity and mortality in immunodeficient patients, and congenital HCMV infection is a leading infectious cause of long-term neurodevelopmental sequelae (*Boppana et al., 2013*). Currently, there is no approved vaccine for CMV, and the existing antiviral drugs show only a modest effect in the treatment of symptomatic congenital cytomegalovirus disease (*Britt, 2017*; *Kimberlin et al., 2015*). To develop new strategies in antiviral therapy and the rational vaccine design, a better understanding of the interplay between the CMVs and the immune system is essential.

The hallmark of CMVs is a large number of viral proteins that interfere with the immune system components in order to suppress host antiviral response. HCMV and mouse CMV (MCMV), the most frequently used model to study the pathogenesis of HCMV infection (*Brizić et al., 2018*; *Reddehase and Lemmermann, 2018*), possess common mechanisms of immune evasion (*Brinkmann et al., 2015*; *Jackson et al., 2017*; *Lemmermann et al., 2012*; *Rölle and Olweus, 2009*; *Brizić et al., 2014*; *Daley-Bauer et al., 2014*). Both CMVs are able to downregulate molecules that have a dual role as NK cell activating and as T cell costimulatory signals (reviewed in *Brizić et al., 2014*). Examples include immunoevasion of natural cytotoxicity receptors (NCRs) and NKG2D (reviewed in *Hudspeth et al., 2013*; *Schmiedel and Mandelboim, 2017*). For instance, it was demonstrated how HCMV proteins UL18 and UL20 downregulate B7-H6, a ligand for the activating NCR, NKp30 (*Charpak-Amikam et al., 2017*). MCMV-encoded m138 affects ligands for NKG2D, as well as B7-1, another potent T cell costimulatory and NK cell activating molecule (*Mintern et al., 2006*; *Lenac et al., 2006*). Recently, we have also shown how MCMV-encoded m20.1 regulates CD155 (PVR) (*Lenac Rovis et al., 2016*) that serves as a ligand for immune receptors DNAM-1, TIGIT and CD96, expressed both on the cells of innate and adaptive immunity. There is no firm consensus how the viral regulation of NK cells affects the activity of T cell population, and a better understanding of viral products that interfere with both the innate and the adaptive arm of immune response is needed.

Clathrin-associated AP-1 complex is a member of the family of heterotetrameric protein complexes that mediate intracellular membrane trafficking in endocytic and secretory transport pathways (*Boehm and Bonifacino, 2001*). In particular, the AP-1 complex is responsible for the polarized sorting and packaging of membrane proteins into clathrin-coated vesicles at the trans-Golgi network (TGN) and/or endosomes (*Ghosh and Kornfeld, 2003*; *Nakatsu et al., 2014*; *Meyer et al., 2000*). The dysfunction of AP-1 complex has gained interest as a viral immunoevasive strategy of human immunodeficiency virus type I (HIV-1), identifying involved HIV proteins and describing their mechanism of action. (*Janvier et al., 2003*). Whether and how cytomegaloviruses evade innate and adaptive immune defenses by hijacking the AP clathrin adaptors is less known.

The m154 protein, belonging to the *m145* gene family of MCMV immunoevasins, is known to regulate cell-surface expression of CD48, a high-affinity ligand for the activating receptor 2B4 (*Zarama et al., 2014*). Here, we demonstrate that m154 downmodulates the surface expression of numerous targets important for NK cell activation and CD8$^+$ T cell costimulation by perturbing the AP-1 sorting and redirecting them to lysosomal degradation. The list includes CD155 (poliovirus receptor, PVR), a protein that has recently emerged as a promising therapeutic target due to its considerable immunoregulatory potential (*Kučan Brlić et al., 2019*) and we show that both HCMV and MCMV induce the accumulation of CD155 in the AP-1 compartment. We identified the motif responsible for the m154 function whose absence results in an attenuated phenotype in vivo. Overall, our results define m154 as a broad-spectrum immunomodulatory protein that interferes with the early NK response along with the virus-specific CD8$^+$ T cell response.

## Results

### MCMV m154 gene product downregulates surface levels of CD155

We have previously shown that MCMV protein m20.1 (*Lenac Rovis et al., 2016*), just like its counterpart, the HCMV protein UL141 (*Tomasec et al., 2005*), retains CD155 in the endoplasmic reticulum (ER) in an immature form, leading to its proteasomal degradation. However, we have also observed that CD155 accumulates outside the ER compartment in MCMV-infected cells (*Figure 1A*, *Figure 1B*, upper panel), irrespective of the ER-resident m20.1 protein (*Figure 1B*, lower panel, *Figure 1—figure supplement 1*). Thus, we aimed to determine if there is an additional MCMV regulator of CD155.

By screening a library of MCMV deletion mutants, we have identified a new viral regulator of CD155 surface expression. *Figure 1C* shows the CD155 surface expression upon infection with MCMV mutants lacking genes of the region m144-m158. Based on the ability of the mutant virus to restore CD155 surface expression, we identified *m154* gene of MCMV as the one responsible for the downregulation of CD155 (*Figure 1C*). We also tested if this downregulation has a functional impact in terms of binding of CD155 to its immune receptors. Infection of cells with WT MCMV led to a

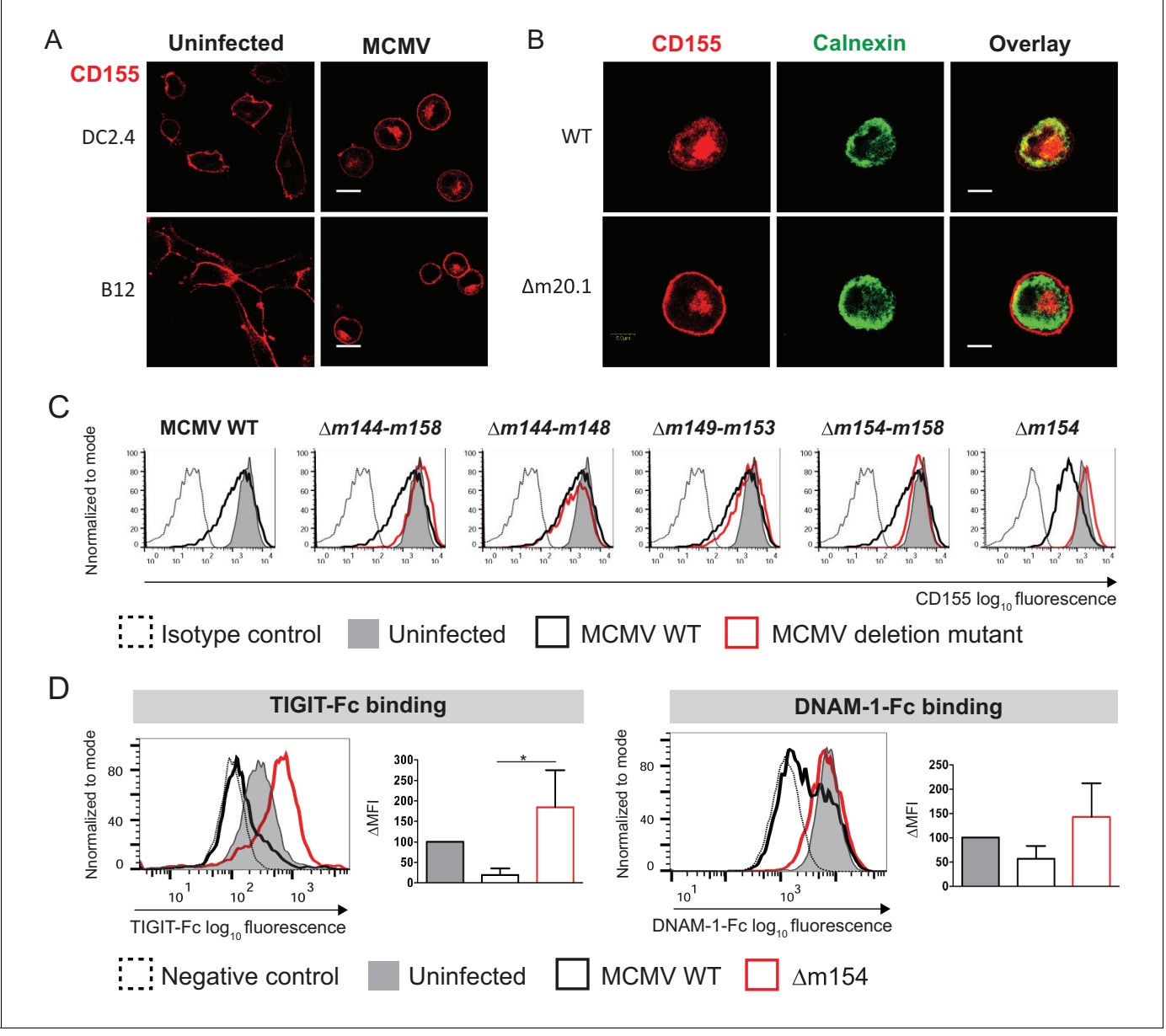

**Figure 1.** MCMV *m154* gene product downregulates surface levels of CD155. (**A**) Confocal images of DC2.4 and B12 mouse cell lines infected with 3 plaque forming units (PFU)/cell of wild-type (WT) MCMV for 20 hr or left uninfected. Cells were stained with an anti-mouse mPVR.01 monoclonal antibody (mAb) followed by anti-rat IgG F(ab')2-TRITC. (**B**) Confocal images of DC2.4 cells infected with Δm20.1 or control WT MCMV as described in (A) or left uninfected. CD155 was stained as described in (A) and endoplasmic reticulum marker calnexin was stained with anti-mouse calnexin followed by anti-rabbit IgG F(ab')2-FITC. For (A and B) scale bar equals 10 μm. (**C**) Flow cytometry analysis of surface CD155 expression on uninfected DC2.4 cells or infected as described in (A) with viral mutants lacking different gene regions or the control WT MCMV. Cells were stained with anti-mouse CD155-PE/Cy7 or isotype control. (**D**) Flow cytometry analysis of TIGIT-Fc and DNAM-1-Fc binding on DC2.4 cells infected with Δm154 or control WT MCMV as described in (A) or left uninfected. Cells were incubated with 2 μg/sample of TIGIT-Fc, DNAM-1-Fc or irrelevant Fc fusion protein, followed by anti-human IgG-FITC. Representative histograms are shown. ΔMFI (difference in median fluorescence intensity) is calculated as 'sample MFI'-'isotype control MFI' and expressed as a percentage of ΔMFI on uninfected cells. Data are representative of at least three independent experiments. Kruskal- Wallis test was used to asses statistical differences with *p<0.05 (p TIGIT-Fc = 0.0158; p DNAM-1-Fc = 0.1051).

The online version of this article includes the following figure supplement(s) for figure 1:

**Figure supplement 1.** MCMV m20.1 protein resides in endoplasmic reticulum.

decrease in TIGIT-Fc and DNAM-1-Fc binding while the viral mutant lacking *m154* gene (Δm154) restored the binding of CD155 interaction partners on the surface of infected cells (*Figure 1D*). The restoration of binding was more pronounced for TIGIT-Fc, since the binding of TIGIT to CD155 is of higher affinity and faster kinetics as compared to DNAM-1 (*Stanietsky et al., 2013*).

## m154 protein and CD155 accumulate in the AP-1 compartment in CMV-infected cells

To determine whether m154 is involved in CD155 intracellular accumulation, we first analyzed its kinetics of expression and localization. In line with the published results (*Zarama et al., 2014*), we observed early kinetics of expression of the m154 protein on the plasma membrane of infected cells.

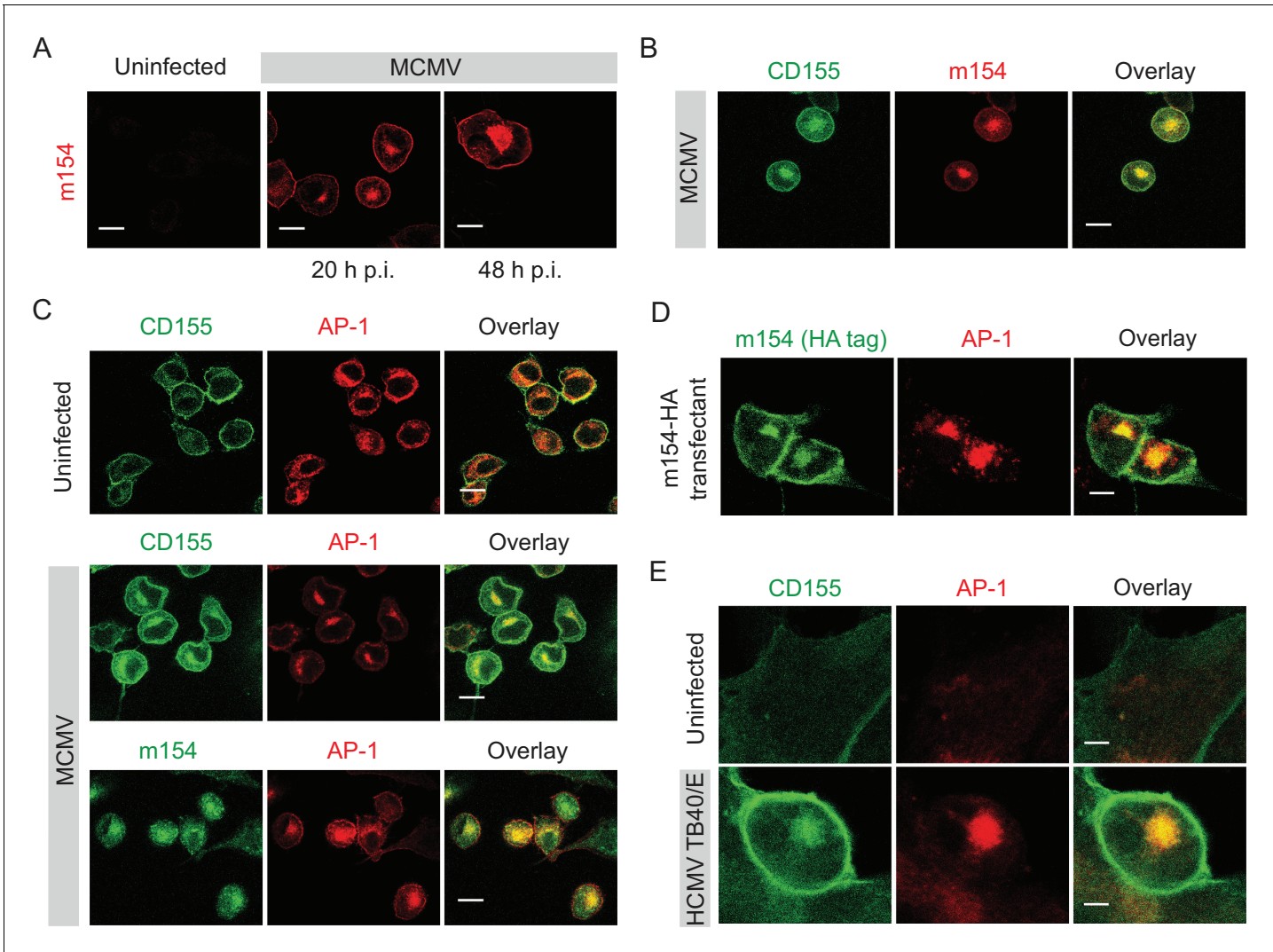

**Figure 2.** m154 protein and CD155 accumulate in the AP-1 compartment in CMV-infected cells. (**A**) Confocal images of DC2.4 cells infected with 3 PFU/cell of WT MCMV. At indicated time points after infection, cells were processed for confocal microscopy and stained with anti-m154 followed by anti-mouse IgG F(ab')2-TRITC. (**B and C**) Confocal images of uninfected DC2.4 cells or infected as described in (A) and stained with anti-mouse mPVR.01 followed by anti-rat IgG F(ab')2-FITC, anti-m154 mAb followed by anti-rat IgG F(ab')2-TRITC or anti-mouse AP-1γ mAb followed by anti-rabbit F(ab')2-TRITC or FITC. (**D**) Confocal images of B12 m154-HA transfectants. Cells were stained with anti-HA followed by anti-rat F(ab')2- FITC, anti-mouse AP-1γ followed by anti-rabbit F(ab')2-TRITC. (**E**) Confocal images of human foreskin fibroblasts (HFF) infected with WT HCMV for 72 hr or left uninfected. Cells were stained with anti-human CD155 followed by anti-mouse F(ab')2-FITC, anti-human AP-1γ followed by anti-rabbit F(ab')2-TRITC. Data are representative of at least two independent experiments. Scale bar in (A, B, C and D) equals 10 μm, and in (E) 5 μm.
The online version of this article includes the following figure supplement(s) for figure 2:

**Figure supplement 1.** CD155 accumulates in AP-1 compartment in cells infected with the virus lacking m154.

However, we also observed that a significant amount of m154 is localized intracellularly (*Figure 2A*). The intracellular expression of m154 overlapped with CD155 intracellular accumulation (*Figure 2B*). After testing a panel of markers for cellular compartments, we determined that in MCMV-infected cells, the accumulated CD155 and m154 colocalize in the adaptor protein-1 (AP-1) positive compartment (*Figure 2C*). The AP-1 protein complex is associated with cargo selection and sorting in trans-Golgi network (TGN), endosomes, and clathrin-coated vesicles (*Ghosh and Kornfeld, 2003*; *Nakatsu et al., 2014*). We also generated stable transfectants expressing m154 N-terminal HA tagged protein and showed that m154 N-HA occupies the expected location in the AP-1 compartment (*Figure 2D*).

Since HCMV also downregulates CD155 surface expression (*Tomasec et al., 2005*), we assessed whether human CD155 colocalizes with AP-1 compartment upon infection. By using human foreskin fibroblasts (HFF) infected with HCMV TB40/E strain, we observed that indeed, human CD155 accumulates intracellularly in the AP-1 compartment (*Figure 2E*). Thus, the AP-1 accumulation of CD155 is a feature of both MCMV and HCMV infection.

## The accumulated CD155 in the AP-1 compartment is derived from the plasma membrane

We showed that in WT MCMV infection m154 localizes to the plasma membrane and AP-1 positive compartment where it also colocalizes with CD155. We examined if the progression of newly synthesized CD155 to the plasma membrane was arrested by m154 on the level of AP-1 compartment. In the cells infected with the virus lacking m154, CD155 was still accumulating in the AP-1 compartment (*Figure 2—figure supplement 1*), indicating that m154 does not directly participate in CD155 retention in this compartment. Next, we investigated the possibility that the accumulated CD155 molecules in AP-1 compartment of MCMV-infected cells were mature, endocytosed CD155 molecules originating from the plasma membrane. To assess this, we stained the plasma membrane CD155 on uninfected and MCMV-infected cells and followed its internalization after 20 hr (*Figure 3A*, schematic model). To avoid the binding of the antibody to the viral Fc receptor, we used the MCMV mutant that lacks *m138* gene encoding for the viral Fc receptor (*Crnković-Mertens et al., 1998*). We observed that CD155 surface staining is quite stable on the plasma membrane of uninfected cells even after 20 hr, while the MCMV-infected cells showed a more pronounced reduction of the initial CD155 surface staining (*Figure 3A*). Using the same surface staining model, but with a permeabilisation step (*Figure 3B*, schematic model), we followed the intracellular trafficking of membrane CD155 by confocal microscopy and found that a significant portion of plasma membrane-stained CD155 (PM-CD155) 20 hr post-infection (p.i.) resided in the AP-1 compartment in MCMV-infected cells, confirming the membrane origin of the intracellularly accumulated CD155 (*Figure 3B*, upper panel). Consistently, the internalized CD155 portion also colocalized with the m154 intracellular expression (*Figure 3B*, lower panel). By using the same surface staining model (*Figure 3C*, schematic model) on m154-N-HA transfectants we showed that surface labeled m154 is also internalized into the AP-1 compartment (*Figure 3C*).

## Plasma membrane CD155 is degraded in lysosomes following MCMV infection

Since the AP-1 clathrin-adaptor is involved in lysosomal enzymes sorting via its TGN-endosome transport function, we next analyzed whether the PM-CD155 molecules of MCMV-infected cells are further processed in lysosomes. To that aim, we stained the PM-CD155, treated cells with lysosomal inhibitor leupeptin and followed the trafficking of internalized CD155 after 20 hr (*Figure 4A*, schematic model). PM-CD155 staining was quite stable on the surface of uninfected cells, regardless of the leupeptin treatment (*Figure 4A*). The MCMV-infected cells showed an increased amount of internalized PM-CD155 after the leupeptin treatment (*Figure 4A*). The rescued portion of PM-CD155 also colocalized with late endosomes and lysosomes, as shown by staining with the acidic organelle probe DAMP. We questioned whether the redirection of PM-CD155 into lysosomes instead of recycling back to the plasma membrane is a direct consequence of m154 action. Δm154 virus possesses viral Fc receptor that would unspecifically bind anti-CD155 antibody in our 20 hr surface staining model. Thus, we labeled plasma membrane proteins with biotin (*Figure 4B*, schematic model) and followed the degradation of the surface portion of CD155 both in WT and Δm154

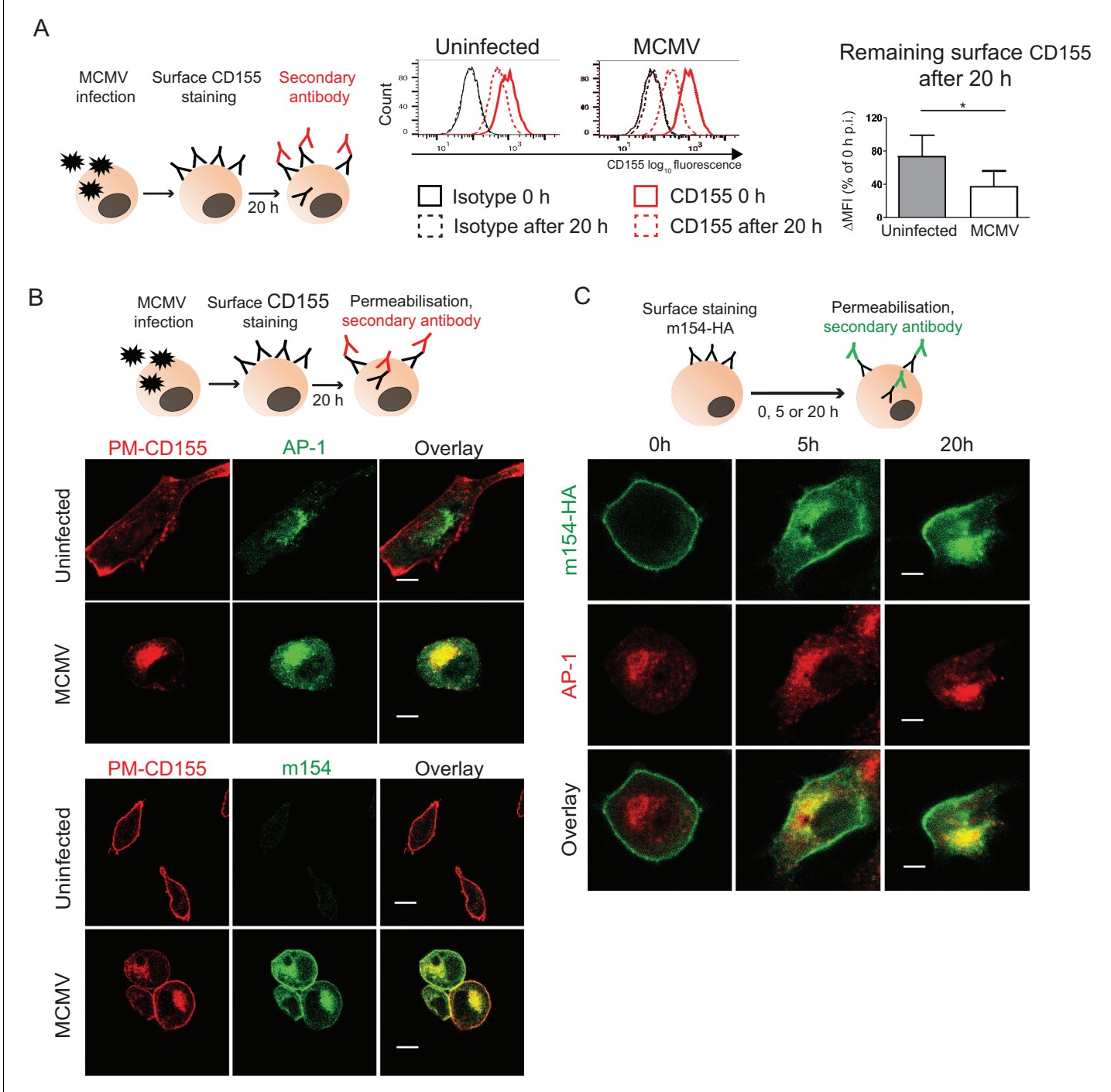

**Figure 3.** The accumulated CD155 in AP-1 compartment is derived from the plasma membrane. (A) Model of surface CD155 staining and later detection. Plasma membrane CD155 (PM-CD155) was stained with anti-mouse mPVR.01 or isotype control on uninfected DC2.4 cells and on the cells that were just infected with 3 PFU/cell of Δm138 virus. After 20 h cells were stained with a secondary antibody anti-rat IgG F(ab')2-FITC and the remaining surface signal was detected by flow cytometry. ΔMFI (difference in median fluorescence intensity) is calculated as 'sample MFI'- 'isotype control MFI' and expressed as a percentage of CD155 signal at 0 hr for the corresponding condition. Two-tailed t test was used to assess statistical differences with *p<0.05 (p=0.022, correlation coefficient = 0.95). (B) Confocal images of DC2.4 cells treated and stained as described in (A) and permeabilized 20 h p.i. In addition, cells were stained with anti-m154 followed by anti-mouse IgG F(ab')2-FITC and anti-mouse AP-1γ mAb followed by anti-rabbit F(ab')2-TRITC or FITC. Data are representative of at least three independent experiments. (C) Model of surface m154-HA staining and later detection. Plasma membrane m154-HA was stained on B12 transfectants with anti-HA mAb. At indicated time points cells were permeabilized and

*Figure 3 continued on next page*

*Figure 3 continued*

stained with secondary antibody anti-rat F(ab')2- FITC. In addition, cells were stained with anti-mouse AP-1γ mAb followed by anti-rabbit F(ab')2-TRITC. Data are representative of at least three independent experiments. Scale bar: 10 μm.

infection by western blot. The biotinylation assay confirmed the loss of CD155 molecules from the plasma membrane of MCMV-infected cells (*Figure 4B*) and the partial rescue of PM-CD155 under leupeptin treatment. More importantly, the degradation of surface CD155 molecules was much less pronounced in the case of Δm154 infection, and there was no further increase of PM-CD155 with

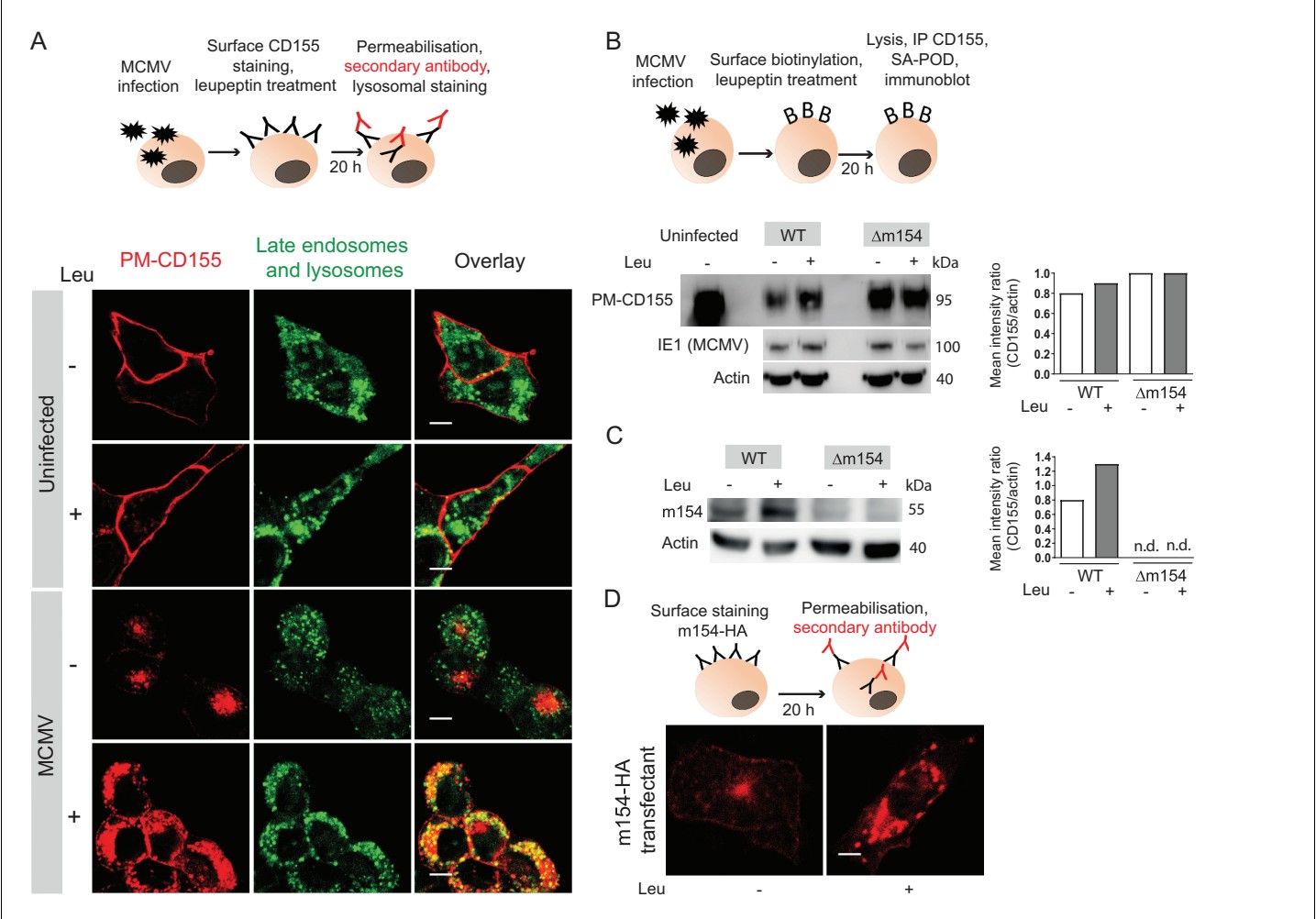

**Figure 4.** Plasma membrane CD155 is degraded in lysosomes following MCMV infection. (**A**) DC2.4 cells were infected with Δm138 virus or left uninfected, and PM-CD155 was stained and tracked as described in *Figure 3*. Four h p.i. cells were treated with leupeptin at 75 μg/ml. 20 h p.i. live cells were stained with the acidic organelle probe DAMP (N-(3-[2,4-dinitrophenyl amino] propyl)-N-(3-aminopropyl)methylamine) for 30 min at 37°C and further processed for confocal microscopy. (**B**) Model of biotinylation of cell surface proteins. B12 cells were biotinylated, infected with 3 PFU/cell of WT MCMV or Δm154 virus or left uninfected. 4 h p.i. cells were treated with lysosomal inhibitor leupeptin and 20 h p.i. cells were lysed, immunoprecipitated with streptavidin and surface CD155 molecules were detected by western blot using mPVR.01 mAb followed by anti-rat IgG-HRP. As a loading control, anti-β-actin mAb was used followed by anti-mouse IgG-HRP, and as a control for infection, lysates were stained with an antibody directed against MCMV protein IE1 followed by anti-mouse IgG-HRP. (**C**) DC2.4 cells were infected with 3 PFU/cell of WT MCMV or Δm154 virus. 6 h p.i. as well as 24 h p.i. cells were treated with lysosomal inhibitor leupeptin and 42 h p.i. cells were lysed. m154 molecules were detected by western blot using anti-m154 mAb followed by anti-mouse IgG-HRP. For (B and C) Individual band intensity was calculated as mean gray value of CD155 or m154 band/mean gray value of the corresponding actin band. (**D**) The plasma membrane m154 was stained with anti-HA mAb on B12-m154-HA transfectant cells, either untreated or treated with leupeptin at 75 μg/ml. After 20 h cells were permeabilized and stained with a secondary antibody anti-rat IgG F(ab')2-FITC and further processed for confocal microscopy. Scale bar 10 μm. Data are representative of at least two independent experiments.

leupeptin treatment. We also demonstrated the involvement of lysosomal enzymes in the degradation of m154 protein in MCMV-infected cells (*Figure 4C*). In addition, PM-m154 accumulated in m154-N-HA transfectants when lysosomal degradation was blocked (*Figure 4D*). Thus, we concluded that m154 interferes with CD155 trafficking by redirecting the PM-CD155 from the AP-1 vesicles to the lysosomal compartment.

## m154 redirects several immunologically relevant targets from the AP-1 compartment to lysosomes

Given the fact that numerous proteins are subjects of AP-1 cargo selection and sorting, we suspected that additional cellular proteins might be affected by the above-described m154 mechanism of action. Therefore, we compared the expression of various surface molecules in cells infected with WT and Δm154 virus and observed an effect of m154 on the surface expression of a remarkable set of molecules involved in innate and adaptive immune responses (*Figure 5A*). This comprised signaling lymphocytic activation family molecules (CD229, CD84), cell adhesion molecules and their receptors (CD18, CD54, CD162), members of tumor necrosis factor receptor family (CD262, CD270), and an integrin-associated protein CD47. The selectivity of m154 downregulation was further shown by examining the surface expression of another set of immunologically relevant molecules that were affected by WT virus but not rescued in the case when m154 gene was missing (*Figure 5B*). We also showed that m20.1, which regulates immature CD155 molecules on the level of ER, has no role in regulating a broad spectrum of targets affected by m154 (*Figure 5—figure supplement 1*).

Next, we wanted to determine whether some of the newly discovered m154 targets follow the same pattern of MCMV-induced sequestration from the cell membrane into the intracellular AP-1 compartment and m154-mediated rerouting for degradation in lysosomes, as observed for CD155. To this end, we used the same model as described in *Figure 3B*, which allowed us to follow the trafficking of surface-stained molecules by confocal microscopy 20 h p.i.. In uninfected cells, the plasma membrane-stained CD229 (PM-CD229) remained quite stable on the cell membrane after 20 hr (*Figure 5C*), while in MCMV-infected cells, the PM-CD229 molecules were internalized and accumulated in the AP-1 compartment (*Figure 5C*). When the lysosomal degradation was inhibited in the infected cells by leupeptin, additional intracellular accumulation of PM-CD229 was detected in infected cells, on the top of its colocalization with the AP-1 compartment (*Figure 5D*). The additional amount of PM-CD229 colocalized with lysosomes in infected cells treated with leupeptin, while the colocalization was minimal when lysosomes were functioning normally (*Figure 5D*). In addition, MCMV induced the accumulation of plasma membrane fraction of other m154 targets, CD18 and CD47, in the AP-1 compartment (*Figure 5—figure supplement 2*).

To conclude, m154 affects the surface expression of multiple targets involved in the antiviral immune response by employing the same mechanism of redirection of the plasma membrane fraction of a protein from the AP-1 compartment to the lysosomal degradation. The AP-1-mediated TGN export of m154 target proteins is blocked by m154, leading to the accumulation of a large fraction of m154 targets in the TGN and a reduction of m154 target proteins on the plasma membrane. A fraction of m154 targets is transported to lysosomes and degraded (*Figure 5—figure supplement 3*).

## The DD motif in the cytoplasmic tail of m154 is necessary for its localization and function

To confirm that AP-1-dependent sorting is involved in the downregulation of m154 targets we silenced AP-1μ subunit by siRNAs. We showed that AP1MI siRNA targeting reduced the ability of MCMV to downregulate CD155 by 50% (*Figure 6A* and *Figure 6—figure supplement 1*). Accordingly, the ability of MCMV to downregulate CD155 was also reduced in cells which do not constitutively express functional AP-1 complex (*Meyer et al., 2000*; *Medigeshi et al., 2008*) (*Figure 6—figure supplement 2*). In the screen of potential functional motifs, the Minimotif Miner database (*Mi et al., 2012*) predicted an AP-binding motif ([DE]D) in the cytoplasmic tail of m154. To investigate the possible involvement of this motif in the protein's function, we have generated a new MCMV mutant (*Figure 6B*) where two aspartic acid residues of the motif (DD) were substituted with two alanine (AA) amino acids (m154-DDAA viral mutant). Currently, the only available antibody that recognizes m154 protein (*Zarama et al., 2014*), was generated by the immunization with a synthetic

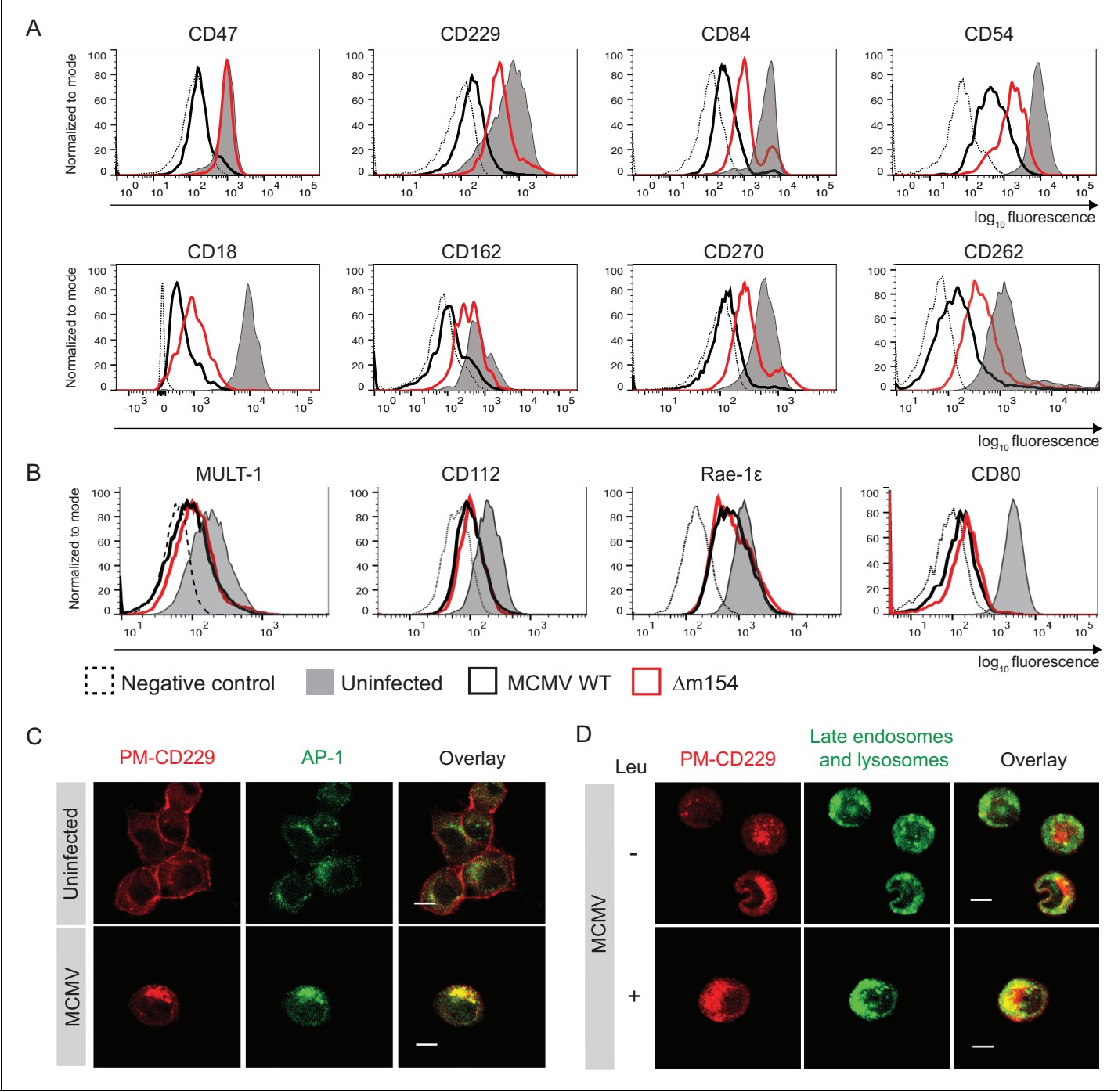

**Figure 5.** m154 redirects several immunologically relevant targets from the AP-1 compartment to lysosomes. (**A**) Flow cytometry analysis of surface molecules of peritoneal macrophages (CD47, CD229, CD84, CD54, CD18, CD162, CD270) and DC2.4 cells (CD262) infected with 10 PFU/cell for 72 hr or 3 PFU/cell for 20 hr, respectively, or left uninfected. (**B**) Flow cytometry analysis of surface expression of MULT-1, CD112, and Rae-1ε on DC2.4 cells infected with 3 PFU/cell of Δm154 or control WT MCMV for 20 hr or left uninfected. For CD80 surface expression, peritoneal macrophages were infected with 10 PFU/cell for 72 hr or left uninfected. (**C**) Plasma membrane CD229 (PM-CD229) on DC2.4 cells was stained with anti-mouse CD229 or isotype control on uninfected DC2.4 cells and on the cells that were just infected with 3 PFU/cell of Δm138 virus. After 20 h cells were processed for confocal microscopy and stained with a secondary antibody anti-mouse IgG F(ab')2-TRITC, and anti-mouse AP-1γ mAb followed by anti-rabbit F(ab')2-FITC. (**D**) DC2.4 cells were infected and stained as described in (**B**) and 4 h p.i. treated with lysosomal inhibitor leupeptin as described previously. 20 h p.i. live cells were stained with lysosomal probe DAMP for 30 min at 37°C and further processed for confocal microscopy. Data are representative of 2 independent experiments. Scale bar: 10 μm.

The online version of this article includes the following figure supplement(s) for figure 5:

*Figure 5 continued on next page*

*Figure 5 continued*

**Figure supplement 1.** MCMV protein m20.1 is not involved in the regulation of surface molecules affected by m154.

**Figure supplement 2.** m154 targets CD18 and CD47 accumulate in the AP-1 compartment upon MCMV infection.

**Figure supplement 3.** Graphical model for the m154 mechanism of action.

peptide corresponding to its cytoplasmic tail. Therefore, to be able to characterize the mutated protein, we introduced an HA tag at the end of the *m154* sequence in m154-DDAA mutant. We have confirmed the expression of the mutated form of m154 in m154-DDAA virus by immunoblotting HA tag in lysates of infected cells (*Figure 6—figure supplement 3A*). The m154-DDAA mutant displayed plaque morphology and in vitro growth kinetics that were indistinguishable from those of the Δm154 and WT virus (*Figure 6—figure supplement 3B*). Next, we showed that m154 protein with a mutated DD motif has reduced ability to localize to the AP-1 compartment in infected cells (*Figure 6C*) and consequently, reduced ability to downregulate the surface expression of its targets (*Figure 6D*). The wild-type m154 with the C-terminal HA tag remained fully functional and downregulated its targets from the surface of infected cells (*Figure 6—figure supplement 4*). In conclusion, we have characterized the DD motif in the cytoplasmic tail of m154 that is responsible for its localization to AP-1 compartment and immunomodulatory function.

## m154 suppresses CD8$^+$ T cell response in vivo

It was previously shown that the viral mutant lacking *m154* is attenuated early after infection in NK-dependent manner, attributing this phenotype to CD48, the first described target of m154 (*Zarama et al., 2014*). Considering that the attenuation was reported for intraperitoneal (i.p.) viral inoculation, we also confirmed the attenuation of Δm154 four days post-infection (p.i.) in an intravenous (i.v.) route of infection (*Figure 7—figure supplement 1A*). Moreover, we observed an even stronger attenuation of Δm154 virus at a later time point of infection (*Figure 7—figure supplement 1B*). Then, we compared the in vivo phenotype of Δm154 and m154-DDAA virus that bears a mutation in the cytoplasmic tail motif. We confirmed the functional role of the DD motif by showing the same levels of attenuation of Δm154 and m154-DDAA viral mutants in different organs and at different times points of infection (*Figure 7A*).

Given the fact that we observed the attenuation of m154-DDAA virus at 7 d p.i. and having showed how the m154 protein downregulates surface molecules involved in CD8$^+$ T cell response, we next determined the role of CD8$^+$ T cells in the control of this virus. While the depletion of NK cells had an effect on the titer of m154-DDAA virus seven days p.i., the additional depletion of CD8$^+$ T cells resulted in a substantial increase of viral titer, supporting the role of CD8$^+$ cell subset in the observed phenotype (*Figure 7B*). Furthermore, these results were comparable to Δm154 infection with NK and CD8$^+$ T cell depletion (*Figure 7—figure supplement 1C*). Of note, CD4$^+$ cell depletion did not further significantly contribute to the observed phenotype (*Figure 7—figure supplement 1C*).

To gain more insight into the role of DD motif in CD8$^+$ T cell response, we next compared CD8$^+$ T cells from WT and m154-DDAA-infected mice. We observed increased frequency and number of MCMV pp89-specific CD8$^+$ T cells in m154-DDAA-infected mice as compared to WT MCMV (*Figure 7C*). To assess the effector function, CD8$^+$ T cells from WT and m154-DDAA-infected mice were stimulated in vitro with MCMV pp89 peptide. Upon 5 hr of stimulation, we detected increased frequency and number of IFN-γ-producing CD8$^+$ T cells in mutant infection compared to WT MCMV (*Figure 7D*) indicating that the DD motif of m154 protein impairs CD8$^+$ T cell response.

To address the effect of m154 on the ability of antigen-presenting cells to induce CD8$^+$ T cell response, we co-cultivated infected bone marrow-derived dendritic cells (BDMCs) with naive CD8$^+$ T cells (*Figure 7E*, schematic model). As a source of CD8$^+$ T cells, we used splenocytes from T cell receptor (TCR)-transgenic mice in which 90% of CD8$^+$ T cells are specific for MCMV M38 peptide (Maxi mice) (*Torti et al., 2011*). m154-DDAA-infected BMDCs induced significantly stronger IFN-γ response by CD8$^+$ T cells as compared to the WT MCMV infection, confirming the role of m154 protein, in particular, its DD motif, in the evasion of CD8$^+$ T cell immune response (*Figure 7E*). All of the indicated findings were also relevant for Δm154 infection (*Figure 7—figure supplement 1D–F*).

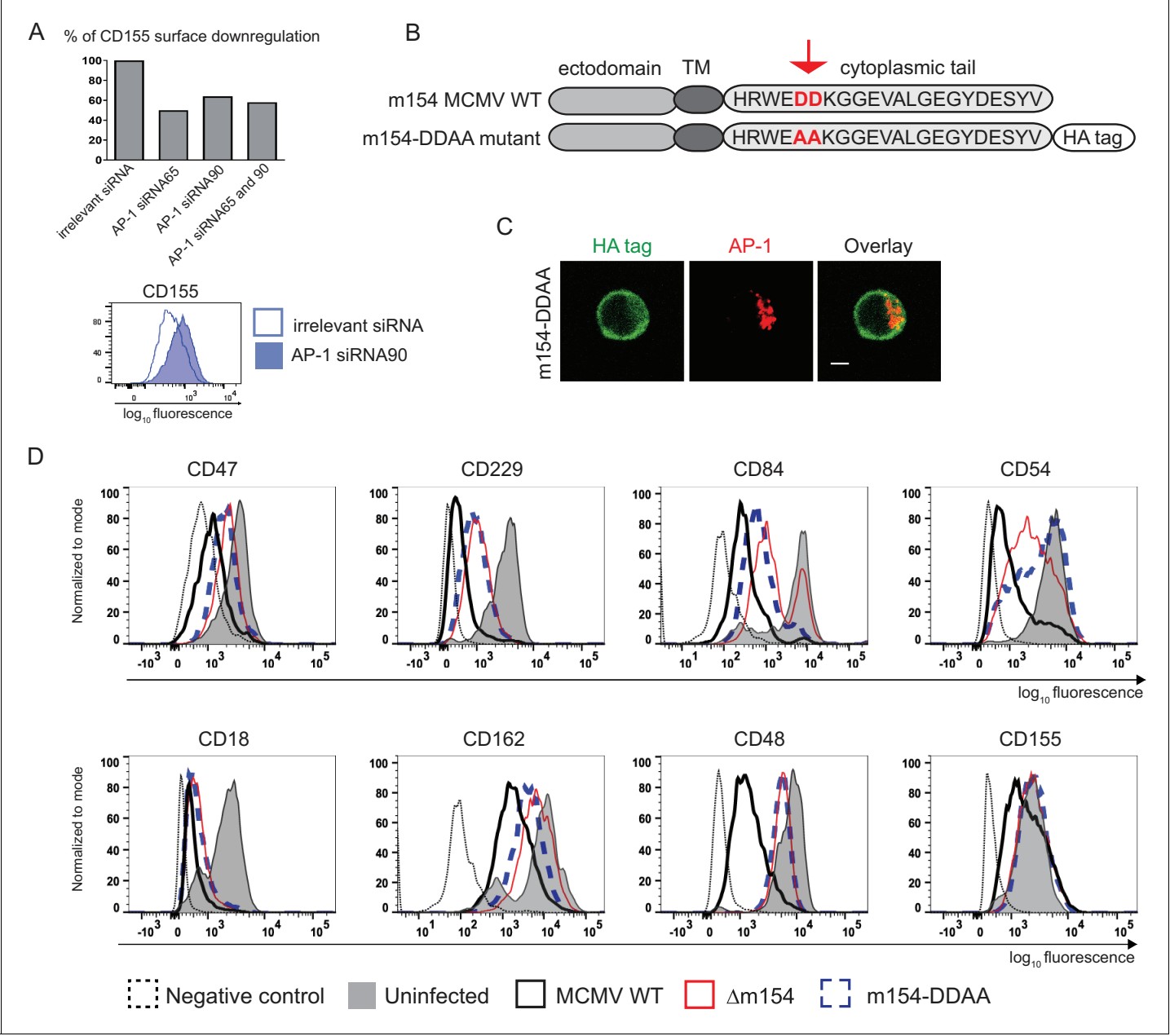

**Figure 6.** The DD motif in the cytoplasmic tail of m154 is necessary for its localization and function. (**A**) Silencing of the AP-1MI gene was done by two independent siRNAs (named 65 and 90) or by their combination in B12 cell line. The surface level of CD155 on WT MCMV- infected AP-1 KD cells was analysed by flow cytometry. Relative % of CD155 downregulation (where irrelevant siRNA treatment represents 100% downregulation) and a representative histogram are shown. (**B**) Schematic representation (not drawn to scale) of m154 protein structure. Two aspartic acid residues in the cytoplasmic tail of WT m154 were substituted by two alanines (AA) (highlighted in red) to generate the MCMV recombinant virus m154-DDAA. TM = transmembrane domain. (**C**) Confocal images of DC2.4 cells infected with 3 PFU/cell m154-DDAA virus for 40 hr and stained with anti-HA followed by anti-rat F(ab')2-FITC, and anti-mouse AP-1γ followed by anti-rabbit F(ab')2-TRITC. Scale bar: 5 μm. (**D**) Flow cytometry analysis of surface molecules of peritoneal macrophages (CD47, CD229, CD84, CD54, CD18, CD162, CD48) and DC2.4 cells (CD155) infected with 10 PFU/cell for 72 hr or 3 PFU/cell for 20 hr, respectively. Data are representative of at least two independent experiments.

The online version of this article includes the following figure supplement(s) for figure 6:

**Figure supplement 1.** siRNA transfection controls.

**Figure supplement 2.** In absence of AP-1 complex, CD155 remains expressed on the membrane of MCMV-infected cells.

**Figure supplement 3.** Characterization of mutated m154 expression and growth kinetics of m154-DDAA viral mutant.

**Figure supplement 4.** Surface expression of CD54 and CD155 is downregulated in infection with MCMV WT m154-HA.

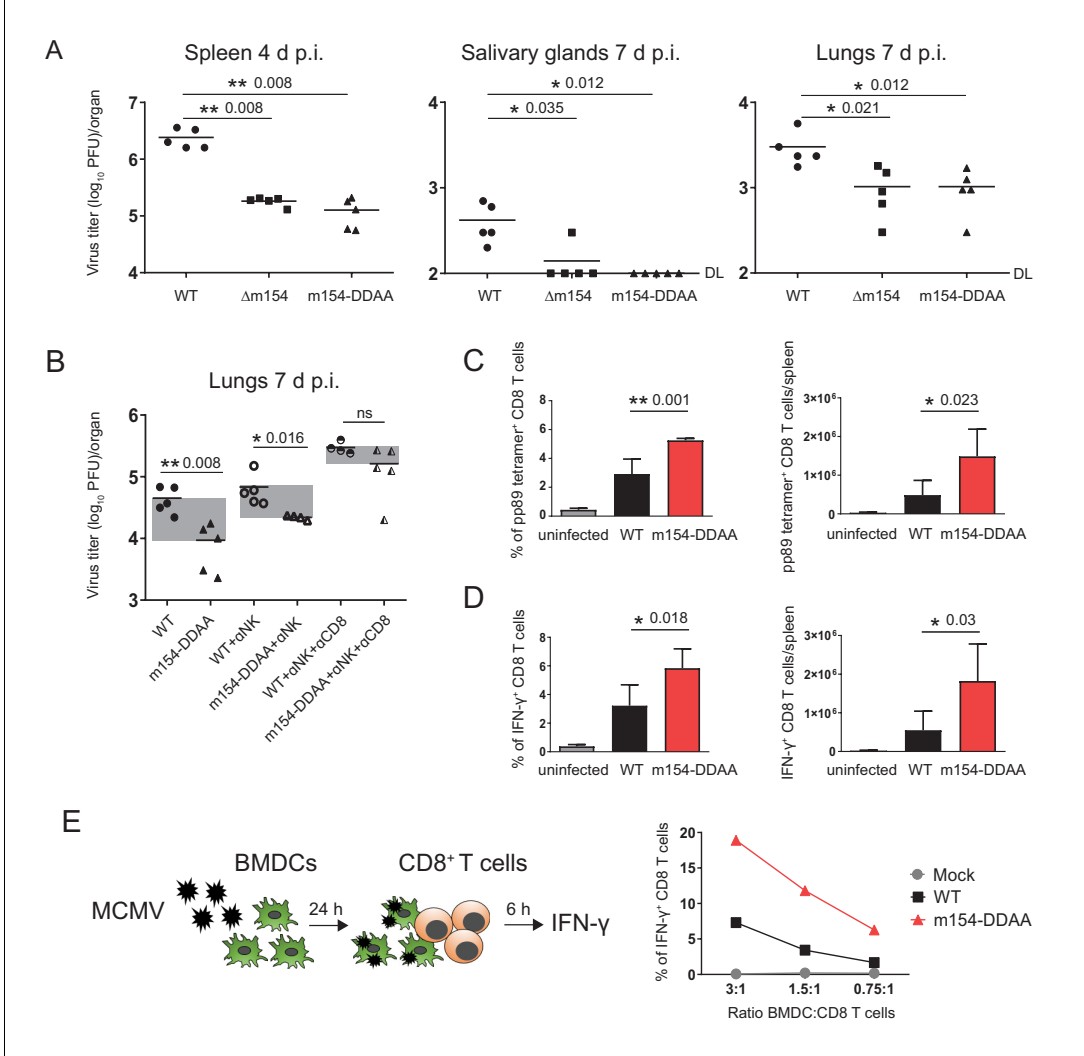

**Figure 7.** m154 suppresses CD8$^+$ T cell response in vivo. (A) BALB/c mice were intravenously (i.v.) injected with $2 \times 10^5$ PFU/mouse of Δm154 MCMV, m154DDAA or WT MCMV as a control. At indicated time points viral titers in organs of individual mice (circles, squares, triangles) were determined by standard plaque assay. Horizontal bars indicate the median values. DL, detection limit. (B) BALB/c mice w/o NK and CD8$^+$ cell depletion were infected as described in (A) and viral titers in lungs were determined 7 d.p.i. by standard plaque assay. For each group of mice in (A and B) n = 5. Two-tailed Mann–Whitney test was used to assess statistical differences between experimental groups. (C and D) Flow cytometry analysis of the percentages and absolute numbers of virus-specific (pp89 tetramer$^+$) and IFN-γ$^+$ splenic CD8$^+$ T cells in mice infected as described in A) and sacrificed 7 d p.i. Two-tailed unpaired t test was used to assess statistical differences between experimental groups. (E) Schematic representation of in vitro antigen-presenting assay. BMDCs were infected with 3 PFU/cell of m154-DDAA mutant or WT MCMV for 24 hr or left uninfected and further co-cultured with naive CD8$^+$ T cells from Maxi mice. After 6 hr, IFN-γ production by CD8$^+$ T cells was determined by flow cytometry. Results are representative of at least two independent experiments. *p<0.05, **p<0.01, ns = not significant. The exact p values are indicated in the figure.

The online version of this article includes the following figure supplement(s) for figure 7:

**Figure supplement 1.** Δm154 exhibits attenuated phenotype in vivo and induces better CD8$^+$ T cell response.

Altogether, we found that m154 impairs CD8$^+$ T cell response and the virus lacking this protein or its functional motif is attenuated in vivo due to the increased capacity to stimulate virus-specific CD8$^+$ T cells.

## Discussion

In this study, we identified m154 as an MCMV protein targeting a large number of immunologically relevant cell surface molecules. The m154 interfered with the sorting of plasma membrane targets at

the level of AP-1 complex and redirected them into the lysosomal compartment. We also showed that productive HCMV infection induces accumulation of human CD155 in the AP-1 compartment, comparable to MCMV-induced accumulation of mouse CD155. We have further identified a DD motif in the cytoplasmic tail of m154, responsible for its AP-1 colocalization and function. The deletion of m154 protein or its DD motif led to the viral attenuation in vivo. With its ability to regulate costimulatory molecules in the AP-1 compartment, m154 reduced the virus-specific CD8$^+$ T cells (*Figure 5—figure supplement 3*).

When it comes to the interference with the AP-1 sorting machinery as a viral strategy to evade immune defense, several viruses have been reported to encode proteins able to interfere with the AP-1 complex, with the Nef protein of HIV-1 being the best-studied example. In particular, Nef stabilizes the association of AP-1 with membranes (*Janvier et al., 2003*) by a mechanism that includes Nef-induced AP-1 trimerisation (*Shen et al., 2015*). The AP-1 trimerisation is cargo-sensitive, directing Nef targets to be either retained intracellularly or sorted to lysosomes (*Morris et al., 2018*). The role of AP sorting machinery in CMV infection was insufficiently investigated so far. In MCMV infection, it was shown that AP-1 and the AP-3 complexes participate in the trafficking of m06/MHC I at the TGN and endosomes respectively (*Reusch et al., 2002*) and that the plasma membrane AP-2 complex mediates endocytosis of the m04/MHC I complexes (*Fink et al., 2015*). Regarding HCMV, the trafficking of UL20 to lysosomes involves the AP-1 complex (*Jelcic et al., 2011*), but no specific cellular targets affected by this action of UL20 were identified so far. Since we showed here the accumulation of CD155 molecules together with AP-1 complexes in the context of productive HCMV infection, it would be interesting to see if UL20 interferes with CD155 expression by this mechanism. Whether CMVs affect AP-1 membrane binding and protein sorting in the same way as HIV-1 (22, 31, 32), involving the direct interaction between the viral protein and the AP-1 complex (*Shen et al., 2015*), remains to be seen. We have successfully immunoprecipitated m154, but our current attempts to co-immunoprecipitate m154 with AP-1 or some of its cellular targets have failed. One of the possible reasons is the lack of suitable antibodies since currently, only one m154 antibody exists (*Zarama et al., 2014*) and it binds exactly the epitope in the cytoplasmic tail of m154, for which we have established to be crucial for the function of m154. Recent findings showed that several alpha herpesviruses also encode proteins able to bind to AP-1, indicating the conserved nature of the described mechanism in herpesviruses family (*Lebrun et al., 2018*). However, no herpesvirus protein has yet been demonstrated to affect multiple targets by interfering with AP-1 mediated protein sorting.

It was recently shown that particular genetic clusters of HCMV regulate a high number of host proteins (*Nightingale et al., 2018*; *Weekes et al., 2014*), but it is not known whether this regulation involves individual or concert action of HCMV genes, nor if this regulation is AP-1-mediated. For MCMV, no single viral protein or genetic cluster with such broad regulatory function has been previously identified. High-throughput approaches, such as mass spectrometry with plasma membrane profiling, could be used to broaden further the extensive list of surface molecules affected by m154. In addition, the absence of a complete rescue of the cell surface levels of the different molecules after infection by Δm154 or m154-DDAA suggests that MCMV might encode for additional viral products that interfere with their expression. For some of the identified m154 targets, such as CD262 (Trail-R2), CD229 (Ly9) or CD162 (P-selectin glycoprotein ligand-1; PSGL-1), it was reported that they are internalized entirely or partially by clathrin-mediated endocytosis (*Austin et al., 2006*; *Lin et al., 2013*; *Del Valle et al., 2003*), but for others, such as GPI-linked glycoproteins, their probability to be affected by the DD motif in the cytoplasmic tail of m154 is less obvious. We have shown the MCMV-induced accumulation of CD18 (β2-integrin) in AP-1-coated membranes, and this integrin was reported to interact with CD54 and CD47, which are also the targets of m154. Therefore, m154 might exert its broad regulating function by affecting molecules with multiple interacting partners such as CD18 integrin. By binding to integrin complexes, CD54 (Intercellular Adhesion Molecule 1; ICAM-1) participates in the initial interactions between T cells and accessory or target cells that precede the formation of immunological synapses (*Kvale and Brandtzaeg, 1993*). CD47, an integrin-associated protein, and CD270, a tumor necrosis factor (TNF) ligand, were shown to costimulate T cell activation (*Reinhold et al., 1997*; *Granger and Rickert, 2003*), and CD229 and CD84 are members of the SLAM family that participate in adhesion reactions between T cells and antigen presenting cells by homophilic interactions (*Romero et al., 2005*; *Martin et al., 2001*). Notably, while CD84 was not reported in our previous study (*Zarama et al., 2014*) to be substantially altered by m154, a

further reassessment identified this SLAM family member as an additional target of the viral protein. Thus, many of the molecules whose putative expression on antigen presenting cells is diminished by m154 are important factors in the activation of antiviral T lymphocyte responses.

It was shown previously that MCMV lacking *m154* is attenuated early after infection in NK-cell dependent manner, and this early attenuation was attributed to the subversion of CD48 signaling (*Zarama et al., 2014*). Importantly, here we showed that in addition to avoiding NK cell control, m154 subverts CD8$^+$ T cell control of infection. We observed an improved CD8$^+$ T cell control of m154-DDAA viral mutant, associated with increased frequency and IFN-γ production of virus-specific CD8$^+$ T cells. By using an in vitro assay, we have shown that m154 impairs the capacity of antigen presenting cells to induce CD8$^+$ T cell responses, suggesting the role of adhesion and costimulatory molecules in the improved CD8$^+$ T cell control of m154-DDAA mutant. Overall, we show that the virus lacking m154 is significantly attenuated in both early and late time points of infection and induces strong antiviral CD8$^+$ T cell response.

After the resolution of acute CMV infection, an immunodominant epitope-specific CD8$^+$ T cell population gradually increases over a long period (*Holtappels et al., 2000*; *Klenerman and Oxenius, 2016*). This characteristic memory inflation makes recombinant CMVs attractive vaccine vectors that induce potent CD8$^+$ T cell memory. Furthermore, many of the CMV immunoevasive genes are non-essential for viral growth opening the possibility for their manipulation and modulation of the antiviral immune response. Our previous studies have highlighted that outstanding vaccine efficacy could be achieved by manipulating non-essential immunoevasive genes (*Trsan et al., 2013*; *Hiršl et al., 2018*; *Tomić et al., 2016*). Based on our results, targeting m154 and its functional counterpart in HCMV could be a promising strategy in further designing of CMV vaccine vectors.

While we have identified the role of NK cells and CD8$^+$ T lymphocytes in the control of MCMV m154 mutant, other immune mechanisms could play a role as well. Namely, the m154 mutant virus remained slightly attenuated even following combined depletion of NK, CD8$^+$ and CD4$^+$ T cells, indicating that other immune control mechanisms may be involved. We have focused our analysis on standard time point (day 7 p.i.) and intravenous route of infection, although we are aware of the fact that other routes of infection could affect the immune response to m154 mutant as well. Importantly, we have identified numerous targets of m154-dependent immunoregulation, some of which could affect the response of other immune cells, such as inflammatory monocytes that could be triggered by CD155 (PVR) (*Lenac Rovis et al., 2016*). However, one cannot exclude additional targets of m154 which could affect immune response.

Altogether, our results define m154 as a broad-spectrum MCMV immunomodulatory protein that prevents the proper sorting function of the AP-1 complex in the trans-Golgi network of the host, leading to the lysosomal degradation and decreased surface expression of several m154 targets implicated in the antiviral response. In addition to NK cell regulation, we uncover that the m154 protein impairs CD8$^+$ T cell response as well. These studies add to the complex mechanism of CMV evasion of CD8$^+$ T cells and indicate an important role of m154 protein. For the efficient vector vaccine development, a better understanding of viral genes like m154 that interfere with both the innate and the adaptive arm of the immune response is of paramount importance.

## Materials and methods

### Cells

B12 (SV40-immortalized BALB/c fibroblasts), MEF (mouse embryonic fibroblasts from BALB/c mice) and HFF (human foreskin fibroblasts) were cultivated in Dulbecco's modified Eagle's medium supplemented with 10% or 3% of fetal calf serum (FCS) and 100 U/ml of penicillin, 100 μg/ml streptomycin and 2 mM L-glutamine. DC2.4 cells (immortalized dendritic cells from C57BL/6 mouse) were cultivated in Roswell Park Memorial Institute (RPMI) 1640 medium supplemented with 10% of FCS and 100 U/ml of penicillin, 100 μg/ml streptomycin and 2 mM L-glutamine, w/o β-mercaptoethanol. To obtain bone marrow-derived dendritic cells (BMDCs), cells from the bone marrow of C57BL/6 mice were cultured for 7 days in RPMI supplemented with 10% of FCS and 20% of supernatant from J558 plasmacytoma cells as a source of GM-CSF cytokine (*Alloatti et al., 2016*). Peritoneal macrophages were elicited from peritoneal exudate cells (PECs) four days following i.p. injection of 1 ml of 3% thioglycollate into BALB/c mice. PECs were collected by peritoneal lavage, plated out at $2 \times 10^5$ cells/

ml in supplemented RPMI 1640 medium and incubated for 2 hr at 37°C, 5% $CO_2$, after which non-adherent cells were washed away with phosphate buffered saline (PBS). HFF, MEF, B12 and DC2.4 have been tested for mycoplasma. There is no information about cell sex for B12 or DC2.4 cell lines available. MEF cells were mixed sex since they were obtained by homogenisation of total embryos 17 days after fertilization. BMDC cells were obtained from female C57BL/6 mice, and PECs from both male and female BALB/c mice. M154-HA fusion protein was constructed as described in Zarama et al. Stable transfectants were obtained by transfecting B12 cells with Extreme Gene nine transfection reagent, according to the manufacturer's protocol. HA-high population was sorted using BD FACSAria II and selection was performed with 0.7 mg/ml of G418. Since μ1A-deficiency is embryonic lethal at embryonic day 13.5, the AP-1μ1A deficient cell lines (here referred AP1-KO) were previously established by continuous passaging of fibroblasts from embryonic day 11.5 mouse embryos (*Meyer et al., 2000*). The AP1-KO cells and the rescued cell line which served as the control cell line (here referred AP1-WT) were cultivated in 10% DMEM without or with hygromicin selection (200 μg/ml) (*Meyer et al., 2000*; *Medigeshi et al., 2008*). All cells were grown in an incubator that enables conditions of 5% $CO_2$ and $37^0$C.

## Viruses

All viruses were propagated on primary BALB/c MEFs and titrated by standard plaque assay as described in *Brizić et al. (2018)*. The BAC pSM3fr-derived MCMV, C3X, based on the MCMV Smith strain (ATCC VR-1399) (*Wagner et al., 1999*) and a recombinant MCMV-GFP (*Mathys et al., 2003*) were used as wild type MCMVs. The generation of the following recombinant MCMV mutants was described elsewhere: mutants which lack different sets of genes from the region *m144-m158* (also called MCMV-Δ6 region) were described in *Brune et al. (2006)*, the Δm138/fcr-1 mutant in *Crnković-Mertens et al. (1998)* and Δm154 and Δm154Int mutant in *Zarama et al. (2014)*. The MCMV mutant m154-DDAA, containing the DD to AA substitution and the sequence encoding HA-tag at the C-terminal portion of the *m154* gene, was constructed by en-passant mutagenesis (*Tischer et al., 2010*), in cells containing pSM3fr BAC. The following primers were used: m154-CTermDelFlag-F: 5'-TCACCGTTGTGATTTTATCTGGGATCGCCGCGGGAGTACTCCTGATCACA TACCCATACGATGTTCCAGATTACGCTTGAAGGATGACGACGATAAGTAGGG-3'; m154-Cterm-DelFlag-R: 5'-AAACACCGCACCAGAGACCAAGTATAAAGCAGTTTTATTGAGCTGATGAG TCAAGCGTAATCTGGAACATCGTATGGGTATGTGATCAGGAGTGTATATCTGGCCCGTACATCGA TCT-3'; m154-CTermDelFlag-short-F: 5'-TCACCGTTGTGATTTTATCT-3'; m154-CtermDelFlag-short-R: 5'-AAACACCGCACCAGAGACCA-3'; Pep-Kan-F: 5'-AGGATGACGACGATAAGTAGGG-3'; Pep-Kan-R: 5'-GTATATCTGGCCCGTACATCGATCT-3'; ORF-Kan-F: 5'-ATGAGCCATATTCAACGGGA-3'; ORF-Kan-R: 5'-CTCATCGAGCATCAAATGAAA-3'; m154-AA-Flag-F: 5'-GATCGCCGCGGGAGTAC TCCTGATCACACACCGTTGGGAAGCAGCAAAGGGTGGGGAGGTGGCACTCGGGGAAGGTTA TGACGAGTCTTATGTGTACCCATACGATGTTCCAGATTACGCTTGAAAGGATGACGACGATAAG TAGGG-3'; m154-AA-Flag-R: 5'- AAACACCGCACCAGAGACCAAGTATAAAGCAGTTTTATTGAGC TGATGAGTCAAGC GTAATCTGGAACATCGTATGGGTACACATAAGACTCGTCATAACC TTCCCCGAGTGCCACCTCCCCACCCTTTGCTGCGTATATCTGGCCCGTACATCGATCT-3'; m154-AA-Flag-F-short: 5'-GATCGCCGCGGGAGTACTCC-3' and m154-AA-Flag-R-short: 5'- AAACACCG-CACCAGAGACCAA-3'. Using analogous approach, a virus encoding HA-tag at the C-terminal portion of the wild-type m154 allele (WT-HA) was constructed using primers: m154-HA-R-long: 5'-AAACACCGCACCAGAGACCAAGTATAAAGCAGTTTTATTGAGCTGATGAGTCAAGCGTAATC TGGAACATCGTATGGGTACACATAAGACTCGTATATCTGGCCCGTACATCGATCT-3', KanF: 5'-AAGGATGACGACGATAAGTAGGG-3', m154-HA-F-long: 5'-ATAAGGGTGGGGAGGTGGCAC TCGGGGAAGGTTATGACGAGTCTTATGTGTACCCATACGATGTTCCAGATTACGCTTGAAAGGA TGACGACGATAAGTAGGG-3' and m154-HA-short-R: 5'-AAACACCGCACCAGAGACCAAG-3'. The accuracy of desired genomic modifications at the target loci in the recombinant viruses was verified by sequencing. Subsequently, all recombinant viruses were reconstituted by transfection of recombinant BAC into primary MEFs, and the expression of the mutated form of the m154 protein encoded by this virus was confirmed by immunoblotting HA-tag from lysates of infected cells. Plaque morphologies and growth kinetics were determined and compared to the parental virus. WT MCMV strains and various MCMV mutants used in this study were propagated on MEF, and virus stocks were prepared as described previously (*Brizić et al., 2018*). Viral titers were determined by standard

plaque assays on MEFs. For infection of HFF cells, we used the HCMV strain TB40/E (*Sinzger et al., 2008*).

## Bacterial strains

*Escherichia coli* strain GS1783 (*Tischer et al., 2010*) was used for the preparation of MCMV BAC pSM3fr (*Wagner et al., 1999*).

## Mice

BALB/c, C57BL/6, and Maxi mice (*Torti et al., 2011*) (MHC-I-restricted TCR-transgenic mice with specificity for the inflationary M38 epitope) were housed and bred under specific-pathogen-free conditions at the Central Animal Facility, Faculty of Medicine, University of Barcelona and Central Animal Facility, Faculty of Medicine, University of Rijeka. All procedures involving animals and their care were approved by the Ethics Committee of the University of Barcelona (protocol number CEEA 308/12) and the Animal Welfare Committee at the University of Rijeka and National ethics committee (Croatia) (Approval reference 525-10/0255-17-4). All procedures were conducted in compliance with institutional guidelines as well as with national (Generalitat de Catalunya decree 214/1997, DOGC 2450) and international (Guide for the Care and Use of Laboratory Animals, National Institutes of Health, 85–23, 1985) laws and policies. Female and male mice aged 8–16 weeks were used in the study.

## Generation of anti-mCD155 monoclonal antibody

The immunogen was a recombinant mouse CD155 protein expressed in Fc mut pIRESpuro vector (*Glasner et al., 2012*) and purified from the supernatant of transfected HEK 293 T cells. The recombinant protein was designed in such a way that the CD155 ectodomain is fused with the human IgG1 Fc fragment (mCD155-Fc). It was purified using prepacked chromatography Protein G HP column (GE) for AKTA systems (AKTA purifier). We used the Dark Agouti (DA) rat strain for immunization. DA rats were injected with 50 μg of the immunogen in complete Freund's adjuvant and 2 weeks later reinjected in incomplete Freund's adjuvant. After 2 weeks, the sera were screened for the antibody titer. The best responders were boosted with the immunogen in PBS. Three days later, spleen cells were collected, and after lysis of red blood cells, fused with SP2/0 cells. The cells were seeded in 20% RPMI 1640 medium containing hypoxanthine, aminopterin and thymidine for hybridoma selection and screened for mAbs production using ELISA. mPVR.01 antibody clone was further selected based on performance in immunofluorescence, flow cytometry and immunoprecipitation methods.

## Determination of virus in vitro growth kinetics

In vitro viral growth was analyzed by infecting MEFs with WT MCMV, Δm154 or m154-DDAA virus at 0.1 PFU/cell. At indicated days p.i. culture supernatants were collected and frozen at −80°C. Further, the amount of extracellular infectious virus present in the culture supernatant was determined by a standard plaque-forming assay (*Brizić et al., 2018*) in two technical replicates.

## Western blot analysis and immunoprecipitation

Cell lysates of DC2.4 cells were prepared using NP-40 lysis buffer (10 mM Sodium phosphate pH 7.2, 150 mM sodium chloride, 2 mM EDTA, 1% NP-40, protease inhibitors). Cells were infected with 3 PFU/cell of WT MCMV or indicated viral mutants. Following infection, cells were supplemented with lysosomal inhibitor leupeptin (75 μg/μl) and lyzed after 20 hr. 75–100 μg of lysate proteins were separated by 10–12% SDS-PAGE electrophoresis and transferred onto 0.45 μm PVDF membranes. Membranes were blocked with 5% w/v nonfat dry milk and incubated with anti-m154 (clone m154.4.113); anti-actin (clone C4), anti-HA Peroxidase (clone 3F10) and anti-MCMV IE1 (clone IE1.01). Protein signals were visualized using UVITec imaging system. For immunoprecipitation of biotinylated mCD155, cells were biotinylated via standard protocol (Thermo Scientific Pierce Cell Surface Protein Isolation/biotinylation), lysates were prepared as above and incubated overnight at 4°C under rotation with anti-mCD155 mAb (clone mPVR.01) followed by 1 hr incubation with protein G sepharose beads (50 μl). The precipitates were washed five times (1 ml each) with IP buffers before the samples were subjected to the western blot analysis. The membranes were incubated

with peroxidase-conjugated streptavidin and visualized as described above. At least two independent experiments were performed for each subject of analysis.

## Immunofluorescence microscopy

B12 and DC2.4 cells were infected with 1–3 PFU/cell of WT MCMV or indicated viral mutants. HFF cells were infected with 1 PFU/cell of HCMV strain TB40/E for 72 hr. In siRNA silencing assays (*Figure 6A* and *Figure 6—figure supplement 1*), short interfering RNAs (siRNAs) pre-designed by Thermofisher: Stealth siRNAs MSS202065 (here referred siRNA65) and MSS273190 (here referred siRNA90) were used to silence adaptor-related protein complex AP-1, mu subunit one gene, alone or in combination. The procedure followed Lipofectamine RNAiMAX reagent protocol with the OPTI MEM transfection medium, LIPOFECTAMINE RNAIMAX transfection reagent, BLOCKIT ALEXA FLUOR RED OLIGO positive control of transfection and STEALTH RNAI NEG CTL HI GC negative control siRNA. In the antibody internalization assays (*Figures 3*, *4* and *5* and *Figure 5—figure supplement 2*), the m154 N-terminal HA transfectant was used, or in the case of infection, the virus lacking *m138* gene (Δm138) that encodes for a viral homolog of Fc receptor was used to exclude the possibility of Fc-antibody binding to the viral Fc receptor during 20 hr of incubation. The equivalence of the WT MCMV and Δm138 virus in terms of specific protein regulation was confirmed in a separate set of analyses. Following infection, cells were treated with leupeptin (75 µg/µl) where indicated and 20 h p.i. fixed and analyzed for target proteins localization. The following in-house produced antibodies were used: anti-mCD155 (clone mPVR.01), anti-hCD155 (clone hPVR.16), anti-MCMV m20.1 (clone m20.1.02), anti-m154 (clone m154.4.113), anti-CD229 (clone 7.144.2). The following purchased antibodies were used: anti-mouse calnexin, anti-mouse γ1-adaptin, anti-HA, anti-mouse CD18 (clone M18/2), anti-mouse CD47 (clone miap301). The matching isotype controls were used. For lysosomal visualization, an acidic compartments probe DAMP (N-(3-[2,4-dinitrophenyl amino] propyl)-N-(3-aminopropyl)methylamine) was used. The samples were stained with secondary antibodies coupled to fluorescein isothiocyanate (FITC) or tetramethylrhodamine (TRITC) fluorophores, mounted using Mowiol mounting medium and analyzed at room temperature with Olympus FV300 confocal laser scanning microscope using Olympus-Microscope-PlanApo-60X-NA1.4-oil objective and FluoView acquisition software without gamma adjustments. At least two independent experiments were performed for each subject of analysis and at least two technical replicates/experiment were analyzed.

## Flow cytometry and cytokine staining

Flow cytometry was performed according to the Guidelines for the use of flow cytometry and cell sorting in immunological studies (*Cossarizza et al., 2017*). Uninfected or cells infected with indicated MCMV strains were stained for cell surface markers: CD155 (clone mPVR.01), CD229 (clone 7.144.2), CD18 (clone M18/2), CD48 (clone HM48-1), CD47 (clone miap301), CD54 (clone 1H4), CD84 (clone mCD84.7), CD162 (clone 2PH1), CD270 (clone HMHV-1B18), CD80 (clone MEM-233), CD262 (clone MD5-1), CD112 (clone 502–57), MULT-1 (clone 1D6), or anti-RAE-1ε mAb (Clone 205001). The matching isotype controls were used. In siRNA silencing assays (*Figure 6A*) we monitored CD155 protein expression on B12 cells following the protocol described in the Immunofluorescence microscopy section. For the viral strains expressing GFP, infected cells were gated as GFP⁺. Otherwise, where needed, infected cells were gated based on the expression of MCMV protein m06 or m04, revealed by staining with anti-m06 (clone croma229) or anti-m04 (clone m04.17). The following secondary antibodies were used: anti-rat IgG F(ab')2-PE or FITC, anti-mouse IgG F(ab')2-PE, FITC, APC or PE-streptavidin. Fixable Viability Dye coupled to Alexa Fluor 780 was used to exclude dead cells. When needed, the blocking with 20% fetal rabbit serum and 1% fetal bovine serum was performed before immunostaining. For IFNγ staining, splenic leukocytes were prepared using standard protocols and incubated in RPMI medium supplemented with 10% of FCS and 1 µg/well of pp89-derived peptide for 5 hr at 37˚C. Brefeldin A at 1 µg/ml was added immediately to the cells. Before staining, Fc receptors were blocked with anti-mouse CD16/CD32 (clone 2.4G2) (*Yokoyama and Kim, 2008*). Subsequently, cells were surface stained, fixed and permeabilized, followed by intracellular staining with anti-IFNγ. The following reagent was obtained through the National Institutes of Health (NIH) Tetramer Core Facility: MCMV-specific tetramer pp89-PE. MCMV pp89-derived peptide was obtained from JPT Peptide Technologies. Flow cytometry data were

acquired on FACSAriaIIu, FACScan (BD Biosciences) or LSRFortessa (BD Biosciences), and analyzed using FlowJo_v10 (Tree Star) software. At least three independent experiments were performed for each subject of analysis.

## Mouse infections and viral titers

For each experiment, mice were pooled and randomized in experimental groups of five animals. Female mice aged 8–12 weeks were i.v. injected with tissue culture-grown recombinant MCMV strains with $2 \times 10^5$ PFU/animal, in a volume of 500 µl of pure DMEM. Depletion of NK cells was performed by intraperitoneal (i.p.) injection of 20 µl of anti-asialo GM1, depletion of CD8+ T cells by i.p. injection of 150 µg of anti-CD8 antibody (YTS 169.4) and depletion of CD4+ T cells by i.p. injection of 150 µg of anti-CD4 antibody (YTS 191.1). Animals were sacrificed at indicated time points, organs were harvested and viral titers were determined by a standard plaque-forming assay (*Brizić et al., 2018*) in two technical replicates. At least two independent experiments were performed for each subject of analysis.

## In vitro antigen presentation assay

Bone marrow-derived dendritic cells (BMDCs) from C57BL/6 mice were obtained as described above and infected with 3 PFU/cell of indicated viruses for 24 hr and used as antigen-presenting cells. Further, they were co-cultured at indicated ratios with splenocytes isolated from Maxi mice during 6 hr in the presence of Brefeldin A at 1 µg/ml. The number of IFNγ+ CD8+ T cells within the total splenocyte population was determined by flow cytometry. At least three independent experiments were performed for each subject of analysis with at least two technical replicates/experiment.

## Statistics

Results are expressed as mean + /- standard deviation of at least three independent experiments. For in vivo experiments, mice were pooled and randomized in experimental groups of five animals. The sample size was determined by pilot studies and according to accepted practice in previous literature utilizing the MCMV model. The selection of the appropriate statistical test was based on the number and distribution of data points per group. Confidence interval was set to 95% and a level of $p < 0.05$ was considered to be statistically significant and assigned as *$p < 0.05$, **$p < 0.001$, ns $p > 0.005$. Exact p values and statistical details of experiments are indicated in the figure legends. Statistical differences between the two study groups were evaluated using paired or unpaired, two-tailed t-test or the non-parametric Mann-Whitney test. Statistical differences between more than two study groups were evaluated using the Kruskal-Wallis bidirectional test. All statistical analyses were performed using GraphPad Prism eight software.

The Key Resources Table (*Supplementary file 1*) lists the reagents and resources used in this study.

## Availability of the materials and resources

Further information and requests of the materials and resources described in the article should be directed to and will be fulfilled by the Correspondence Contact, Tihana Lenac Rovis (tihana.lenac@medri.uniri.hr) or Stipan Jonjic (stipan.jonjic@medri.uniri.hr).

## Acknowledgements

We thank Suzana Malic, Karmela Miklic, Edvard Razic, and Dijana Rumora for excellent technical assistance in biochemical methods, monoclonal antibody production and virus production. We also thank Miro Samsa, Edvard Marinovic, Ante Mise and Maja Cokaric Brdovcak for assistance in breeding and handling animals. We thank Annette Oxenius (Institute of Microbiology, ETH Zürich, Zürich, Switzerland) for providing the Maxi mice. The following reagents were obtained through the National Institutes of Health (NIH) Tetramer Core Facility: MCMV-specific tetramer pp89-PE, MCMV pp89-derived peptide. Guillem Angulo was supported by a fellowship 'Formación de Personal Investigador' from the Ministerio de Economía y Competitividad (MINECO, Spain). Funding Acquisition, AA (Ministerio de Economía y Competitividad (MINECO, Spain) grant no. SAF2017-87688), TLR (Croatian Science Foundation grant no. UIP-11-2013-1533; University of Rijeka grant no. uniri-

biomed-18–23), SJ ("Strengthening the capacity of CerVirVac for research in virus immunology and vaccinology", grant no. KK.01.1.1.01.0006, awarded to the Scientific Centre of Excellence for Virus Immunology and Vaccines and co-financed by the European Regional Development Fund). The funding sources were not involved in study design, data collection and interpretation, or the decision to submit the work for publication

## Additional information

### Competing interests
Stipan Jonjic: Reviewing editor, *eLife*. The other authors declare that no competing interests exist.

### Funding

| Funder | Grant reference number | Author |
| --- | --- | --- |
| Ministerio de Economía y Competitividad | SAF2017-87688 | Ana Angulo |
| Hrvatska Zaklada za Znanost | UIP-11-2013-1533 | Tihana Lenac Rovis |
| European Regional Development Fund | KK.01.1.1.01.0006 | Stipan Jonjic |

The funders had no role in study design, data collection and interpretation, or the decision to submit the work for publication.

### Author contributions

Ivana Strazic Geljic, Paola Kucan Brlic, Data curation, Formal analysis, Investigation, Visualization; Guillem Angulo, Data curation, Formal analysis, Investigation; Ilija Brizic, Berislav Lisnic, Tina Jenus, Vanda Juranic Lisnic, Methodology; Gian Pietro Pietri, Data curation, Investigation; Pablo Engel, Resources; Noa Kaynan, Investigation; Jelena Zeleznjak, Formal analysis, Methodology; Peter Schu, Data curation; Ofer Mandelboim, Conceptualization; Astrid Krmpotic, Conceptualization, Methodology; Ana Angulo, Conceptualization, Resources, Supervision, Funding acquisition, Project administration; Stipan Jonjic, Conceptualization, Resources, Supervision, Funding acquisition; Tihana Lenac Rovis, Conceptualization, Resources, Supervision, Funding acquisition, Visualization, Methodology, Project administration

### Author ORCIDs

Ivana Strazic Geljic https://orcid.org/0000-0001-9394-493X
Paola Kucan Brlic https://orcid.org/0000-0002-9178-9128
Ofer Mandelboim http://orcid.org/0000-0002-9354-1855
Tihana Lenac Rovis https://orcid.org/0000-0002-3299-1334

### Ethics
Animal experimentation: All procedures involving animals and their care were approved by the Ethics Committee of the University of Barcelona (protocol number CEEA 308/12) and the Animal Welfare Committee at the University of Rijeka and National ethics committee (Croatia) (Approval reference 525-10/0255-17-4). All procedures were conducted in compliance with institutional guidelines as well as with national (Generalitat de Catalunya decree 214/1997, DOGC 2450) and international (Guide for the Care and Use of Laboratory Animals, National Institutes of Health, 85-23, 1985) laws and policies.

### Decision letter and Author response
Decision letter https://doi.org/10.7554/eLife.50803.sa1
Author response https://doi.org/10.7554/eLife.50803.sa2

## Additional files

### Supplementary files
- Supplementary file 1. Key Resources Table.
- Transparent reporting form

### Data availability
All data generated or analysed during this study are included in the manuscript and supporting files.

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
