## [Decision Letter]

**Acceptance summary:**

The manuscript shows a role for the mouse CMV (MCMV) protein m154 in immune evasion via modulation of AP-1-dependent intracellular trafficking of several co-stimulatory molecules. The authors have shown previously that m154 mediates degradation of CD48, a high-affinity ligand for the natural killer and cytotoxic T cell receptor CD244, and have shown that an MCMV mutant lacking m154 is attenuated in vivo, and that this can be partially rescued when NK cells are depleted. They now show that m154 reduces cell surface expression of several other proteins, particularly CD155, mediating lysosomal degradation by interference with AP-1-dependent trafficking, and identify the motif in m154 responsible for this function. They also show some evidence that human CMV (HCMV) also leads to accumulation of CD155 in an AP-1-positive compartment. Overall, this study focuses on the MCMV m154 protein which disturbs trafficking of cellular proteins, and thereby confers an important advantage for successful evasion of host immune defence.

**Decision letter after peer review:**

Thank you for submitting your article "Cytomegalovirus protein m154 perturbs the adaptor protein-1 compartment mediating broad-spectrum immune evasion" for consideration by *eLife*. Your article has been reviewed by two peer reviewers, and the evaluation has been overseen by Satyajit Rath as the Senior and Reviewing Editor. The reviewers have opted to remain anonymous.

The reviewers have discussed the reviews with one another and the Reviewing Editor has drafted this decision to help you prepare a revised submission.

Essential revisions:

1) It should be tested whether m154 binds to any of its identified substrates, e.g. by IP/blot in leupeptin-treated cells. Is the m154 on the plasma membrane described previously a separate pool to that in the AP-1 compartment? Thus, if you surface label m154, does it get internalised with similar kinetics to its degradation substrates? Is it also degraded in lysosomes?

2) Subsection “m154 redirects several immunologically relevant targets from the AP-1 compartment to lysosomes”: it would seem important to show that a wild-type m154 with a C-terminal HA tag is fully functional – not just expressed – since this is the tag they put on their DDAA mutant protein which shows a loss of function. Currently, like is not clearly compared with like.

3) The authors haven't clearly shown how the DD mutant affects AP-1 – they demonstrate that a proposed AP-1 binding mutant is no longer localised in an AP-1 positive compartment. However, this does not prove that it is a result of a lack of AP-1 binding. This is acknowledged in the Discussion, but much of the text assumes that they have identified a binding motif – which is not proven. It should be possible to show that an AP-1 KO or KD loses the effect of m154 on its targets?

4) It is hard to know what to make of the CD8 response measurements. Given cross-priming and all the other evasions in operation, the effect is surprisingly large. Is the effect CD8-specific? An effect on NK cell (or inflammatory macrophage) recognition could feed through to dendritic cells and T cells. It is necessary to show that another immune response is unaffected by m154 disruption, e.g. CD4^+^ T cells. The CD48 effect of m154 was attributed to NK cell evasion. There is good precedent for evading both NK and CD8; e.g. m152 was initially a CD8 evader, then later an NK evader. It is not convincing to compare the degree of m154 reversion by CD8 and NK depletions; the data are not very tight, and the depletions differ in efficacy. No depletion shows full reversion and other innate killers, such as inflammatory monocytes, may also turn out to be involved. Given that the targeted receptors are shared across cell types, expecting simple conclusions is too much. One suspects also that changing the mouse strain, MCMV strain or infection route could give a different hierarchy. Probably most of this could be dealt with by toning down conclusions and acknowledging uncertainty.

5) The emerging model needs to be better expressed and explained both graphically and in text.

[Editors' note: further revisions were requested prior to acceptance, as described below.]

Thank you for resubmitting your work entitled "Cytomegalovirus protein m154 perturbs the adaptor protein-1 compartment mediating broad-spectrum immune evasion" for further consideration by *eLife*. Your revised article has been evaluated by Satyajit Rath, Senior and Reviewing Editor.

The manuscript has been improved but there are some remaining issues that need to be addressed before acceptance, as outlined below:

1) In Figure 6—figure supplement 2, the authors seem to show that in AP-1 KO (AP‐1μ1A-deficient) cells the CD155 is downregulated upon MCMV infection while there is no effect in the AP-1 WT control cells. However, in the text, they write “We also demonstrated loss of m154-mediated regulation of CD155 protein in cells that constitutively lack the AP1 μ1-adaptin (Meyer et al., 2000; Medigeshi et al., 2008) subunit Figure 6—figure supplement 2)”. Could this be clarified, please?

2) The explanatory figure (Figure 5—figure supplement 3. Graphical model for m154 mechanism of action) simply shows a generalised cartoon based on a generic view of the secretory pathway. It would be much more useful to have a figure with a more detailed explanation of how the authors think m154 affects AP1-mediated trafficking of their target proteins, and the effect of the DD/AA motif.

---

## [Author Response]

Essential revisions:1) It should be tested whether m154 binds to any of its identified substrates, e.g. by IP/blot in leupeptin-treated cells. Is the m154 on the plasma membrane described previously a separate pool to that in the AP-1 compartment? Thus, if you surface label m154, does it get internalised with similar kinetics to its degradation substrates? Is it also degraded in lysosomes?

The only anti-m154 antibody currently available is directed against the intracellular portion of the m154 protein. Therefore, to allow surface monitoring of m154 internalization, we now produced stable transfectants expressing m154 as N-terminal HA tagged protein. Our data showed that m154 N-HA is localized on the plasma membrane and in the AP-1 compartment (new Figure 2D). Using these transfectants we were able to demonstrate that surface labeled m154 is internalized into the AP-1 compartment and follows similar internalization kinetics as its degradation substrates (new Figure 3C). We have also shown that internalized m154 accumulates when lysosomal degradation is blocked (new Figure 4D) and the effect of lysosomes on the degradation of the m154 protein was also demonstrated in infected cells (new Figure 4C). Although we repeatedly tried to co-immunoprecipitate m154 with its degradation targets from leupeptin-treated cells, we did not obtain results that would confidently demonstrate interaction employing this method. We think there could be two reasons for this. First, the fact that m154 and its targets accumulate in lysosomes, which cannot degrade them due to leupeptin, does not mean that under lysosomal-compartment conditions they still form a stable complex, if such a complex otherwise exists. Second, a bioinformatic analysis of the m154 protein sequence revealed that a large part of the ectodomain sequence (more than 50%) represents Low-Complexity Regions which are regions known to allow specific protein flexibility and adaptation to different interacting partners based on their positions within a sequence. Hence, harsh conditions of the immunoprecipitation method could disrupt regions necessary for m154 interaction.

2) Subsection “m154 redirects several immunologically relevant targets from the AP-1 compartment to lysosomes”: it would seem important to show that a wild-type m154 with a C-terminal HA tag is fully functional – not just expressed – since this is the tag they put on their DDAA mutant protein which shows a loss of function. Currently, like is not clearly compared with like.

The only currently available antibody against m154 was obtained by immunization with a synthetic peptide corresponding to the intracellular tail portion of m154, which contains the amino acids DD, and therefore the antibody does not recognize the mutated m154 protein (m154-DDAA). Thus, when creating the m154-DDAA virus, we had to add the HA tag to its cytoplasmic tail. We agree with the reviewer that the control wild type m154-HA virus (WT-m154-HA) needs to be proven functional. To address this issue, we constructed a new virus in which the HA tag was added to the C terminus of the intact m154 encoding gene. In the new Figure 6—figure supplement 4 we show now that the WT-m154-HA virus expresses the HA-tagged m154 protein. More importantly, this virus downregulates the CD155 protein on B12 cells, whereas the m154-DDAA mutant virus is unable to do it. We show the same phenomenon for CD54 protein on DC2.4 cells. This information has also been included in the new Figure 6—figure supplement 4 of the revised manuscript.

3) The authors haven't clearly shown how the DD mutant affects AP-1 – they demonstrate that a proposed AP-1 binding mutant is no longer localised in an AP-1 positive compartment. However, this does not prove that it is a result of a lack of AP-1 binding. This is acknowledged in the Discussion, but much of the text assumes that they have identified a binding motif – which is not proven. It should be possible to show that an AP-1 KO or KD loses the effect of m154 on its targets?

We thank the reviewers for their suggestions. We performed the experiments to answer these questions and we confirmed that AP-1 KO or KD loses the effect of m154 on its targets. We used previously published siRNA molecules that suppress the expression of the µ subunit of the AP-1 protein (AP-1 KD) and a cell line that does not constitutively express the AP-1 protein (AP-1 KO). The AP-1 KO cell line was previously obtained from embryonic cells of the AP-1 KO mouse, since it represents a lethal mutation (Meyer et al., 2000). The efficacy of siRNA transfection in AP-1 KD cells was controlled by i) the manufacturer's positive fluorescence control (new Figure 6—figure supplement 1) and ii) the AP-1γ subunit staining, since the AP1 protein lacking the µ subunit cannot form functional complex and cannot correctly localize the AP-1 protein in the TGN region (new Figure 6—figure supplement 1). Upon MCMV infection, the ability of the virus to reduce the surface expression of CD155 protein was decreased in AP-1 KD cells (new Figure 6A). Also, the ability of the m154 protein to affect the regulation of the CD155 was diminished in the case of AP1-KO cells (new Figure 6—figure supplement 2). Nevertheless, we agree with the reviewer that this is not a direct evidence of the interaction of the DD motif in the m154 protein with the AP-1 adapter protein, and therefore in the new version of the manuscript we avoided using the term ‘The AP-binding ([DE]D) motif’. For example, we replaced this expression with: ‘identified a DD motif in the cytoplasmic tail of m154, responsible for its AP-1 localization and function’.

4) It is hard to know what to make of the CD8 response measurements. Given cross-priming and all the other evasions in operation, the effect is surprisingly large. Is the effect CD8-specific? An effect on NK cell (or inflammatory macrophage) recognition could feed through to dendritic cells and T cells. It is necessary to show that another immune response is unaffected by m154 disruption, e.g. CD4^+^ T cells. The CD48 effect of m154 was attributed to NK cell evasion. There is good precedent for evading both NK and CD8; e.g. m152 was initially a CD8 evader, then later an NK evader. It is not convincing to compare the degree of m154 reversion by CD8 and NK depletions; the data are not very tight, and the depletions differ in efficacy. No depletion shows full reversion and other innate killers, such as inflammatory monocytes, may also turn out to be involved. Given that the targeted receptors are shared across cell types, expecting simple conclusions is too much. One suspects also that changing the mouse strain, MCMV strain or infection route could give a different hierarchy. Probably most of this could be dealt with by toning down conclusions and acknowledging uncertainty.

We thank the reviewers for the comments on the in vivo results of the study. We agree that the effects of m154 regulation on the expression of numerous molecules involved in the regulation of immune response must have a complex outcome. In addition, as pointed out by the reviewers, this outcome may vary with different experimental conditions, such as route of infection, tissue analyzed or mouse strain used. We also agree that other immune cells are probably involved in the control of m154 mutant. Therefore, as suggested, we have attenuated statements on the role of CD8^+^ T cells in the manuscript, and emphasized that we have found an important role for CD8^+^ T cells in control of this mutant virus, while leaving open the possibility of the role of other cell types. Also, we have conducted a number of new experiments that confirm and complement our main results and which we present here, following the comments of the reviewers:

It is hard to know what to make of the CD8 response measurements. Given cross-priming and all the other evasions in operation, the effect is surprisingly large. Is the effect CD8-specific? An effect on NK cell (or inflammatory macrophage) recognition could feed through to dendritic cells and T cells. It is necessary to show that another immune response is unaffected by m154 disruption, e.g. CD4^+^ T cells.

To verify the results of the CD8^+^ T cell response, we have used Ly49H-/- C57BL/6 mice. Similarly as in BALB/c mice (Figure 7 and Figure 7—figure supplement 1), we have observed stronger CD8^+^ T cell responses in terms of higher frequency of M45- and M57-tetramer positive CD8^+^ T cells in mice infected with the m154 mutant virus, while the frequency of M38-positive cells was the same as in WT MCMV infected mice (Author response image 1, three panels on the left). These results confirm our observations presented in the manuscript. In our previous work we reported that purified NK cells show enhanced responses (enhanced degranulation) to m154 deficient MCMV as compared to WT MCMV infected cells, arguing for the direct recognition of infected cells by NK cells (Zarama et al., 2014).

Furthermore, we have analyzed CD4^+^ T cell responses to m154-deficient MCMV. To that aim we have performed adoptive transfer of TCR-transgenic CD4^+^ T cells specific for M25 protein (M25-II; Mandaric et al. PLoS Pathog 2012) into Ly49H-/- C57BL/6 mice. We observed similar frequencies of M25-II CD4^+^ T cells in mice infected with m154 mutant and WT MCMV (Author response image 1, right panel). These data argue against a role of m154 in evasion of CD4^+^ T cells. However, since we have analyzed only one epitope of CD4, we cannot claim that this is the case with other epitopes.

The CD48 effect of m154 was attributed to NK cell evasion. There is good precedent for evading both NK and CD8; e.g. m152 was initially a CD8 evader, then later an NK evader. It is not convincing to compare the degree of m154 reversion by CD8 and NK depletions; the data are not very tight, and the depletions differ in efficacy. No depletion shows full reversion and other innate killers, such as inflammatory monocytes, may also turn out to be involved.Given that the targeted receptors are shared across cell types, expecting simple conclusions is too much. One suspects also that changing the mouse strain, MCMV strain or infection route could give a different hierarchy.

We agree with the reviewers that the degree of reversion is not an ideal way to show the role of CD8^+^ T cells. However, by comparing different forms of presenting data and applying different combinations of depletion, we have concluded that the way of data presenting that we use is the most informative. Importantly, the results were consistent between experiments and the same was observed with both m154 MCMV mutants (m154-DDAA MCMV and Δm154 MCMV) (Figure 7 and Figure 7—figure supplement 1). Furthermore, similar dependence on CD8^+^ T cell control was observed also in C57BL/6 mice (Author response image 1). It is important to emphasize that C57BL/6 mice are inherently resistant to MCMV infection due to NK cell control of infection as compared to MCMV sensitive BALB/c mice, which were mostly used in our study. Therefore, C57BL/6 mice that were not depleted of NK cells exhibited virus levels that were very low or under detection limit in both WT MCMV and m154-DDAA MCMV infected groups.

We agree with the reviewers that other immune cells could play a role in the control of the m154 MCMV mutant. When we tested m154 mutant virus in NSG (NOD SCID γ) mice, we observed a minor, but significant, attenuation of the m154 mutant virus (Author response image 1). Since NSG mice lack the majority of immune cells, this opens a possibility for a role of other mechanisms in the control of the m154 mutant virus.

**Author response image 1. respfig1:** Effect of m154 distruption on immune control mediated by other immune cell populations. (**A**) 10^5^ of M25-II TCR-transgenic CD4^+^ T cells (CD45.1+) were adoptively transferred into Ly49H-/- C57BL/6 mice (CD45.2+). One day following adoptive transfer mice were i.v. infected with 2x10^5^ PFU of WT MCMV or m154-DDAA mutant. 7 days after infection the frequency of virus-specific splenic CD8^+^ T cells was determined by tetramer staining (left panels; M45, M38 and M57) and flow cytometry. The frequency of M25-II positive cells was determined by analysis of CD45.1 expression (right panel). (**B**) C57BL/6 mice were infected with 2x10^5^ PFU of WT MCMV or m154-DDAA mutant. NK and CD8^+^ T cells were depleted where indicated by the use of monoclonal antibodies. On day 7 p.i. viral titers in lungs of individual mice (circles, squares) were determined by standard plaque assay. (**C**) Balb/c and NSG mice were infected with 2x10^5^ PFU of WT MCMV or Δm154 mutant. On day 7 p.i. viral titers were determined as described in B. Horizontal bars indicate the median values. DL, detection limit. Two-tailed Mann-Whitney test was used to assess statistical differences between experimental groups. For each group of mice in B) n=6 and in C) n=5.

5) The emerging model needs to be better expressed and explained both graphically and in text.

We have included now in the new version of the manuscript a graphical model for the m154 mechanism of action (new Figure 5—figure supplement 3) and we have also added a more detailed textual explanation.

[Editors' note: further revisions were requested prior to acceptance, as described below.]

The manuscript has been improved but there are some remaining issues that need to be addressed before acceptance, as outlined below:1) In Figure 6—figure supplement 2, the authors seem to show that in AP-1 KO (AP‐1μ1A-deficient) cells the CD155 is downregulated upon MCMV infection while there is no effect in the AP-1 WT control cells. However, in the text, they write “We also demonstrated loss of m154-mediated regulation of CD155 protein in cells that constitutively lack the AP1 μ1-adaptin (Meyer et al., 2000; Medigeshi et al., 2008) subunit Figure 6—figure supplement 2)”. Could this be clarified, please?

We apologize for the confusion in the text related to Figure 6—figure supplement 2. In the new revised version of the manuscript, we modified confusing sentence and revised the respective figure to emphasize the message we wish to deliver.

2) The explanatory figure (Figure 5—figure supplement 3. Graphical model for m154 mechanism of action) simply shows a generalised cartoon based on a generic view of the secretory pathway. It would be much more useful to have a figure with a more detailed explanation of how the authors think m154 affects AP1-mediated trafficking of their target proteins, and the effect of the DD/AA motif.

According to suggestions, we tried to improve the graphical model for m154 mechanism of action described in Figure 5—figure supplement 3.